# A Novel Nanobody Precisely Visualizes Phosphorylated Histone H2AX in Living Cancer Cells under Drug-Induced Replication Stress

**DOI:** 10.3390/cancers13133317

**Published:** 2021-07-01

**Authors:** Eric Moeglin, Dominique Desplancq, Audrey Stoessel, Christian Massute, Jeremy Ranniger, Alastair G. McEwen, Gabrielle Zeder-Lutz, Mustapha Oulad-Abdelghani, Manuela Chiper, Pierre Lafaye, Barbara Di Ventura, Pascal Didier, Arnaud Poterszman, Etienne Weiss

**Affiliations:** 1Biotechnologie et Signalisation Cellulaire, UMR 7242, CNRS/Université de Strasbourg, Boulevard S. Brant, 67412 Illkirch, France; eric.moeglin@unistra.fr (E.M.); dominique.desplancq@unistra.fr (D.D.); audrey.stoessel@unistra.fr (A.S.); gabrielle.zederlutz@unistra.fr (G.Z.-L.); chiper@unistra.fr (M.C.); 2Signaling Research Centers BIOSS and CIBSS and Institute of Biology II, University of Freiburg, Schänzlestrasse 1, 79104 Freiburg, Germany; christian.massute@web.de (C.M.); jeremy.ranninger@neptun.uni-freiburg.de (J.R.); barbara.diventura@biologie.uni-freiburg.de (B.D.V.); 3Institut de Génétique et de Biologie Moléculaire et Cellulaire Illkirch CEDEX, 67404 Illkirch, France; alastair.mceven@igbmc.fr (A.G.M.); oulad@igbmc.fr (M.O.-A.); arnaud.poterszman@igbmc.fr (A.P.); 4Centre National de la Recherche Scientifique (CNRS), UMR 7104, 67404 Illkirch, France; 5Institut National de la Santé et de la Recherche Médicale (INSERM), U1258, 67404 Illkirch, France; 6Université de Strasbourg, 67404 Illkirch, France; 7Plateforme d’ingénierie des Anticorps, C2RT, UMR 3528, CNRS/Institut Pasteur, Rue du Dr. Roux, 75015 Paris, France; pierre.lafaye@pasteur.fr; 8Laboratoire de Bioimagerie et Pathologies, UMR 7021, Faculté de Pharmacie, CNRS/Université de Strasbourg, 74 Route du Rhin, 67401 Illkirch, France; pascal.didier@unistra.fr

**Keywords:** H2AX, phosphorylation, replication stress, nanobody, imaging, one-step detection, cancer cells, genotoxicity assay in live cells

## Abstract

**Simple Summary:**

γ-H2AX, a phosphorylated variant of histone H2A, is a widely used biomarker of DNA replication stress. To develop an immunological probe able to detect and track γ-H2AX in live cancer cells, we have isolated single domain antibodies (called nanobodies) that are easily expressed as functional recombinant proteins and here we report the extensive characterization of a novel nanobody that specifically recognizes γ-H2AX. The interaction of this nanobody with the C-terminal end of γ-H2AX was determined by X-ray crystallography. Moreover, the generation of a bivalent nanobody allowed us to precisely detect γ-H2AX foci in drug-treated cells as efficiently as with commercially available conventional antibodies. Furthermore, we tracked γ-H2AX foci in live cells upon intracellular delivery of the bivalent nanobody fused to the red fluorescent protein dTomato, making, consequently, this new cost-effective reagent useful for studying drug-induced replication stress in both fixed and living cancer cells.

**Abstract:**

Histone H2AX phosphorylated at serine 139 (γ-H2AX) is a hallmark of DNA damage, signaling the presence of DNA double-strand breaks and global replication stress in mammalian cells. While γ-H2AX can be visualized with antibodies in fixed cells, its detection in living cells was so far not possible. Here, we used immune libraries and phage display to isolate nanobodies that specifically bind to γ-H2AX. We solved the crystal structure of the most soluble nanobody in complex with the phosphopeptide corresponding to the C-terminus of γ-H2AX and show the atomic constituents behind its specificity. We engineered a bivalent version of this nanobody and show that bivalency is essential to quantitatively visualize γ-H2AX in fixed drug-treated cells. After labelling with a chemical fluorophore, we were able to detect γ-H2AX in a single-step assay with the same sensitivity as with validated antibodies. Moreover, we produced fluorescent nanobody-dTomato fusion proteins and applied a transduction strategy to visualize with precision γ-H2AX foci present in intact living cells following drug treatment. Together, this novel tool allows performing fast screenings of genotoxic drugs and enables to study the dynamics of this particular chromatin modification in individual cancer cells under a variety of conditions.

## 1. Introduction

Histones constitute the core proteins of chromatin and their post-translational modifications (PTMs) contribute to the molecular basis of epigenetic gene regulation and cellular memory [1]. In humans, several variant forms of histones have been described [2] and this is particularly relevant for the H2A histone. The H2A variants represent the largest and most diverse family of histones; there is overwhelming evidence that their unstructured N- and C-termini, which protrude out of the core structure of the nucleosome, harbor several sites for PTMs in response to varying stimuli [3]. The H2AX variant shares high amino acid similarity with H2A and is characterized by an extended C-terminus, which is phosphorylated when the cells become injured by agents that provoke DNA replication stress (RS) and genome instability [4]. The phosphorylation of serine at position 139 (S139) of H2AX has been particularly well studied and represents a key event in the detection and response to DNA damage [5,6].

Phosphorylation of histone H2AX at S139, which gives rise to what is generally referred to as γ-H2AX, is in fact a very early step in the DNA damage response (DDR) and an essential signal for the recruitment and retention of DDR complexes at the site of damage [7]. Three different phosphatidylinositol 3 kinase (PI3K)-related kinases mediate S139 phosphorylation on H2AX: ATM (ataxia-telangiectasia mutated), ATR (ATM and Rad3-related), and DNA-PK (DNA-dependent protein kinase) [8]. ATM and DNA-PK share functional redundancy upon ionizing radiation, while ATR may preferentially phosphorylate H2AX during RS [9]. This PTM of H2AX is highly dynamic and a number of phosphatases, including those of the PPP family and Wip1, are able to dephosphorylate γ-H2AX to fine-tune the duration and intensity of the DDR signaling [10]. It has also been found that H2AX can be phosphorylated at the threonine residue at position 136 (T136) and at the C-terminal tyrosine residue at position 142 (Y142) to facilitate DNA repair, whereas the persistency of the latter PTM may also trigger apoptosis [11]. Nevertheless, S139 phosphorylation is regarded as the main PTM of H2AX since it is specifically recognized by the adaptor protein MDC1, which further recruits several E3 ubiquitin ligases to favor DNA repair and/or restart of the halted forks during RS [12].

Because γ-H2AX is involved in the DDR, it is generally considered a biomarker of DNA double-strand breaks (DSBs) [13] and its relevance as read-out of sustained RS is well accepted [14]. In addition, H2AX is also phosphorylated in the absence of DNA breakage, likely during replication fork arrest and subsequent single-stranded DNA accumulation, and this early event upon insult induces the formation of discrete nuclear foci of γ-H2AX, which can be visualized with specific antibodies under the microscope. The formation of γ-H2AX, which can spread progressively over the whole nucleus (pan-nuclear γ-H2AX) following chromatin modification by loop extrusion [15], gives in fact an estimate of the severity of the RS [16]. γ-H2AX is considered, nowadays, a universal bio-indicator of the severity of genotoxic compounds that interfere with DNA replication in vitro and in vivo [17]. Indeed, almost all studies aiming at selecting small molecules triggering irreversible genome instability refer to γ-H2AX formation and retention to assess their potency. In particular, tracking γ-H2AX is of high interest for validating chemotherapeutics [18] and for controlling the carcinogenic properties of chemicals present in biological samples [19]. Immunofluorescence with validated antibodies remains so far the method of choice for accurately determining γ-H2AX levels [20]; however, currently, there is no simple tool available for monitoring γ-H2AX turn-over and for measuring its direct impact on cell viability.

In a previous study, we have shown that delivery in cells of antigen-binding fragments (Fabs) derived from an anti-γ-H2AX monoclonal antibody (mAb) allows the fate of cancer cells after treatment with varying RS-inducing drug combinations to be followed [21]. Although it was possible to show with this method that extensive γ-H2AX generation is indicative of commitment to irreversible cell death, we could not clearly identify the dynamic changes of γ-H2AX formation during the treatment. Indeed, conventional antibodies cannot be easily delivered in the nucleus of living cells, and, therefore, require fixation of the samples.

Recently, it has been shown that single-domain antibody fragments of camelids, generally termed nanobodies, represent exquisite tools for tracking intracellular molecules [22,23]. Nanobodies correspond to the variable domain (VHH) of the heavy chain-only antibodies (HcAb) expressed in these animals [24]. They can be cloned as VHH repertoires with minimal modification from total RNA of peripheral blood mononuclear cells (PBMCs) obtained after immunization, thus, presenting an authentic picture of the in vivo-maturated heavy chain repertoire diversity. Moreover, their small size (~15 kDa) compared to conventional antibodies (~150 kDa) and, especially, their capacity to fold stably in a reducing environment make them excellent binding molecules in cells [22,23,24,25]. While alpaca-derived nanobodies against γ-H2AX have already been generated [26], these tools did not allow for the specific unambiguous detection of γ-H2AX in irradiated cancer cells.

Here, we developed a novel nanobody with high specificity against γ-H2AX. We demonstrate that bivalency is essential to achieve high specificity against the antigen. We solved the crystal structure of the nanobody in complex with the phosphopeptide representing the C-terminus of H2AX, which enabled us to design mutations that impair the binding. We show the ability of this novel nanobody to quantitatively visualize γ-H2AX in fixed as well as in live cells with the same precision as a validated mAb. To analyze γ-H2AX in real time, we developed a transduction strategy based on the delivery of nanobody-dTomato fusion proteins by electroporation, allowing its efficient transport in the crowded nucleus and precise binding to γ-H2AX foci in this compartment upon drug treatment. The novel nanobody presented here will make it possible to study the dynamics of H2AX phosphorylation at S139 in living cells in unprecedented ways. The transduction approach is versatile and can easily be implemented with other nanobodies, allowing a new way to visualize proteins and their PTMs inside cells with high specificity.

## 2. Results

### 2.1. Development and Selection of Specific Anti-γ-H2AX Nanobodies by Phage Display

To produce a nanobody to track human γ-H2AX in cells under RS, we immunized alpacas with the phosphorylated peptide CKATQA(p)SQEY corresponding to the C-terminal end of γ-H2AX (residues 134–142). This peptide has been used in a previous study to generate monoclonal antibodies that are suitable for detecting γ-H2AX in various immunoassays [16]. Following immunization, the PBMCs were collected and VHH libraries of approximately 10^7^ independent clones were constructed. The phage display technology, which consists in displaying the VHH molecules on the tip of M13-based phages, allows selecting those that bind to the phosphopeptide immobilized on plate. This method of antigen display was preferred to other methods, such as immobilization on magnetic beads, since it previously allowed successful screening of cell culture supernatants containing monoclonal antibodies [16]. Colony counting following the first round of panning (R1) showed that phages expressing a VHH against the phosphopeptide were only present in the repertoire of one animal (library 2) (Figure 1A). The same results were obtained after the second round of selection, indicating that only a low number of the cloned VHH molecules bound to the antigen. This suggests that immunization with the phosphopeptide did not trigger a strong heavy chain-only antibody response in the animals, but promoted mainly the synthesis of IgG1, since the sera collected from the three animals were positive when tested by ELISA with anti-alpaca immunoglobulins. Importantly, anti-γ-H2AX heavy chain-only antibodies could not be detected in the sera by immunoprecipitation with peptide-coated beads.

We tested the binding specificity of the VHH-displaying phages selected from library 2 by comparing their reactivity against the phosphorylated and non-phosphorylated peptides coated on plate (phage-ELISA). The selected populations bound preferentially to the phosphopeptide, which suggests that the phosphate group at S139 is important for the recognition (Figure 1B, left). Interestingly, we also observed that phages corresponding to the library without selection (R0) were able to react with the phosphopeptide, albeit to a lesser extent (Figure 1B). Similar results were obtained when the same experiment was performed with nuclear extracts of U2OS cells treated for 24 h with hydroxyurea (H), which induces RS and causes H2AX phosphorylation at S139 (Figure 1B, right). This suggests that the VHH molecules displayed on the surface of the selected phages recognize γ-H2AX.

Individual clones of the positive phage population (R1 and R2 from library 2) were amplified and subjected to DNA sequencing. The alignment of 165 different sequences showed that almost all analyzed VHH clones have a similar amino acid composition, except at positions 98 and 99 (Kabat numbering) in the complementary-determining region 3 (CDR3) (Figure 1D), meaning that they all may have arisen from a single B cell. The specific binding of the four most-represented variants (A4, A9, C6, and G2) to the phosphopeptide was confirmed by phage-ELISA (Figure 1C). Collectively, these results show that the screening by phage display of several immune VHH libraries led to the selection of a unique VHH scaffold that specifically binds to γ-H2AX.

### 2.2. The Selected Nanobodies Are Soluble in the Bacterial Cytoplasm

To test whether the four identified VHH variants could be used as nanobodies in immunoassays and cells, we first sub-cloned their coding regions into a bacterial vector equipped with the relevant tags for detection and purification, then expressed them as single polypeptides in the cytoplasm of *E. coli* cells. SDS-PAGE analysis of cell extracts showed that the four nanobodies behave differently, despite their high amino acid sequence homology: C6 and A9 are soluble, whereas A4 and G2 are mostly insoluble after cytoplasmic expression (Appendix A). The yield of the purified A9 and C6 nanobodies, which migrate as single bands on gel (Figure 2A), was in the range of 8–20 mg/L of bacterial culture. Their capacity to bind to the phosphopeptide was tested by ELISA. Both reacted with the antigen when used at a concentration of 0.1–10 µg/mL (Figure 2B). No reactivity towards the non-phosphorylated peptide was observed under these conditions (Figure 2B), confirming the results obtained with the phage-ELISA. We then tested the performance of the purified A9 and C6 nanobodies in immunofluorescence. Both showed the typical staining of γ-H2AX following treatment of H1299 cells with hydroxyurea or with a combination of gemcitabine and the Chk-1 inhibitor AZD-7762 (Figure 2C and Appendix A). Treating the cells with both drugs induces intense RS [16]. Interestingly, image quantification indicated that the background staining in this assay was always lower with C6 than that with A9 (Figure 2D). Despite their ability to detect the phosphorylated C-terminus of H2AX in fixed cells, these nanobodies could hardly reveal γ-H2AX foci, which were instead readily observed with the well characterized anti-γ-H2AX mAb 3F4 [16] (Appendix A). Overall, the results suggested that both C6 and A9 can be solubly expressed at high yields in bacteria and, therefore, represent valuable tools for γ-H2AX detection. However, they cannot be considered as equivalent to conventional antibodies.

### 2.3. 3D-Structure Determination of the C6 Nanobody

To understand the interaction between the nanobodies and the phosphopeptide at the atomic level, we solved the crystal structure of the complex at 1.8 Å resolution. We selected the C6 nanobody due to its higher stability upon storage and overall better performance compared to A9. The crystals belonged to space group P3_1_, with six equivalent copies of the complex in the asymmetric unit where significant electron density is observed for the last five residues of the peptide (Figure 2E; Appendix A). The other residues are highly flexible or disordered, implying that they are not involved in specific interactions. The nanobody adopts a canonical IgG fold with a scaffold of nine antiparallel β-strands forming two sandwiching β–sheets [28]. The paratope accepting the phosphopeptide is mainly built from CDR2 and CDR3 resulting in a solvent accessible surface area buried in the interface of approximately 385 Å^2^. Detailed analysis of the complex showed that the phosphate group of phospho-S139 makes direct water-mediated interactions with side chains from CDR2 and CDR3 (Figure 2F). Key residues (single letter code) that belong to CDR2 are the hydrogen bond donors T52, S53, and T56 as well as R55, which also provides an electrostatic contribution. Interestingly, the nanobody interacts also with the last two residues of the peptide (Figure 2G). This second binding pocket involves side chains from CDR3 with key roles for R100, R100C, and R100D: the ammonium group of R100 is stacked against the aromatic ring of the Y142 tail, while those of R100C and R100D recognize the side chain of E141 and the carboxy-terminal group of the phosphopeptide, respectively. Thus, the phosphate group of the phosphopeptide is a crucial determinant of the recognition of the antigen by the C6 nanobody, explaining its exquisite specificity for the modified peptide.

### 2.4. The C6 and A9 Nanobodies Are Solubly Expressed in Mammalian Cells

Next, we examined the behavior of the C6 nanobody when expressed in mammalian cells. We cloned the coding region of C6 fused in frame to mCherry to generate a chromobody [29] expressed under the control of the β-actin promoter and transiently transfected it into H1299 cells. The C6 chromobody was located in the nucleus of the treated cells and to our surprise also to the untreated cells after analysis with either a widefield (Figure 3A) or a confocal microscope (Appendix A). We speculated that unspecific binding to a nuclear antigen was caused by the overexpression of the chromobody. To discern specific from unspecific binding, we treated the cells with cytoskeletal (CSK) buffer prior to fixation. This treatment allows washing out all soluble (unbound and/or weakly bound) proteins while retaining the interactions of stably bound material [30]. Under such stringent conditions, the signal of the chromobody was lost in both drug-treated and untreated cells (Figure 3A), indicating that its nuclear localization upon transfection does not correspond to specific antigen binding and that the affinity of C6 for γ-H2AX may not be sufficient to counteract the CSK wash. The same results were obtained when C6 and mCherry were substituted with A9 and GFP, respectively (Appendix A). Thus, despite their solubility in mammalian cells, these reagents cannot be transfected into cells to detect with precision γ-H2AX after drug treatment likely due to too high chromobody expression levels.

### 2.5. Behavior of the C6 Nanobody Following Transduction

In a previous work, we have shown that antibodies and fragments thereof can be efficiently transduced into cultured cells by electroporation [21,30]. Given their small size (15–20 kDa), nanobodies can theoretically easily diffuse into the nucleus after delivery in the cytoplasm. Therefore, we transduced the purified C6 nanobody in H1299 cells subsequently treated with H and imaged them after 24 or 48 h of incubation. As shown in Figure 3B, the fluorescent signal resembled that typically observed for γ-H2AX, although background staining (without treatment) was also significant. Since a similar staining was observed with the transduced Fab prepared by papain digestion of mAb 3F4 [16] (Figure 3B), we concluded that the C6 nanobody binds to γ-H2AX under physiological incubation conditions. Nevertheless, the background signal of C6 was above that obtained with Fab 3F4 (Figure 3B, right panels), which may indicate that the recognition of the antigen under these conditions is likely less stable for C6 than for Fab 3F4. Since our aim was to develop a nanobody that can be used in live cancer cells at low concentrations, we decided to further improve the binding affinity of the C6 nanobody.

### 2.6. The Bivalent C6 Nanobody Allows Highly Accurate Detection of γ-H2AX in Fixed Drug-Treated Cells

It has been previously reported that the affinity of a nanobody for its antigen can be improved by increasing the valency, for instance genetically connecting two copies of the VHH domain with a linker region to generate a so-called bivalent nanobody [31]. Thus, we constructed a bivalent C6 nanobody (called C6B hereafter) and a mutated version of it (called C6BM), where the two R residues at positions 100C and 100D, which are critical for binding (see Figure 2G), were replaced with alanine and isoleucine, respectively (Figure 4A). Both constructs were expressed in *E. coli* cells and, after purification and validation on gel (Appendix A), their capacity to bind to the phosphopeptide immobilized on plate was tested by ELISA (Appendix A). This experiment showed that C6B bound specifically to the phosphopeptide, whereas C6BM was, as expected, no longer reacting.

To assess whether the purified C6B molecules harbor two functional binding sites, we performed quantitative binding assays using the surface plasmon resonance (SPR) technology. To calculate the affinity of C6 for the antigen (K_D_ value) we used purified monovalent molecules and found that it lies in the low nanomolar range (11 +/− 4 nM; Appendix A). To compare the binding properties of the C6 and C6B nanobodies, saturating amounts were injected separately over a surface coated with an equal amount of the phosphopeptide. The responses were normalized to the peptide density and to the nanobody molecular weight, which allows the calculation of the fractional occupancy (FO) of the peptide sites [32]. At saturation, an FO of one is expected for a 1:1 antibody-antigen molar ratio, while an FO of 0.5 is expected for a homogenous bivalent binding (i.e., 1:2 antibody-antigen molar ratio). As shown in Figure 4B, the FO of C6B bound to the phosphopeptide was significantly reduced compared to that obtained with C6, indicating that a large proportion of C6B interacts with the antigen in a bivalent manner. These results, in addition to the slower dissociation rate of C6B observed on the sensorgrams (Figure 4B), strongly suggest that both VHHs comprising C6B are able to bind to the immobilized phosphopeptide.

To examine if this property of binding by avidity represents an advantage for the detection of γ-H2AX in drug-treated H1299 cells, we performed IF experiments as done for the monovalent molecules (Figure 4C). In this case, upon treatment of the cells with hydroxyurea, γ-H2AX foci could be distinguished more clearly than with the monovalent nanobody (compare Figure 2C and Figure 4C). The same result was obtained with hydroxyurea-treated U2OS cells (Appendix A), indicating that the bivalent nanobody allows detecting γ-H2AX with high precision in fixed cells. No signal was observed with the C6BM mutant nanobody (Figure 4C, right panel), which confirms that the staining observed with C6B represents specific binding. Importantly, the staining obtained with C6B when used at a concentration of approximately 2 ng/mL (Appendix A) is identical to that observed with mAb 3F4 (Appendix A), suggesting that bivalent binding at low concentration is necessary to visualize discrete amounts of γ-H2AX in cells.

### 2.7. Single-Step Detection of γ-H2AX in Fixed Drug-Treated H1299 Cells

To test if C6B could be used as a single-step reagent to detect γ-H2AX foci in fixed cells, we added a cysteine residue in the coding region of C6B between the C-terminus of the second VHH and the E6 tag. The purified protein (C6BC) was labelled with Alexa-Fluor 568-maleimide and used in IF (Figure 4D). γ-H2AX foci could be distinctly detected and quantified with the fluorescently-labelled C6BC molecules when using different combinations of RS-inducing drugs used in the clinic (Figure 4E). When these results were compared to those obtained with the 3F4 mAb used for the screening of the drugs under similar conditions, a linear correlation was obtained with a Pearson correlation coefficient of 0.966 (Figure 4F) demonstrating that fluorescently-labelled C6BC performs as well as the validated mAb for detecting γ-H2AX foci in fixed cancer cells.

### 2.8. The Transduced Bivalent C6 Nanobody Allows Monitoring γ-H2AX in Drug-Treated Live Cells

To investigate whether C6B could be used in cells, we modified the previously constructed chromobody C6-mCherry to add a second VHH copy, thus, creating C6B-mCherry. Upon transfection of H1299 cells with this construct, a strong nuclear mCherry signal was observed (Appendix A). In contrast to what was observed with IF, nuclear staining was also observed in the absence of drug treatment, indicating a certain degree of unspecific binding. Nonetheless, CSK treatment showed that a large fraction of the fluorescent signal remained in the nucleus (Appendix A). No signal was detectable after CSK treatment in cells transfected with the mutant C6BM-mCherry construct (Appendix A). These results, together with the fact that the monovalent C6-mCherry molecules were entirely washed off from the nucleus upon CSK treatment (Figure 3A), clearly indicate that bivalency is of importance for observing binding to the antigen in cells. However, in transient transfection conditions when the plasmid-borne chromobody is highly expressed in cells, unspecific binding remains an issue.

Next, we investigated the performance of the bivalent C6B and C6BM nanobodies by transduction since the corresponding polypeptides showed single bands on gel after purification (Appendix A). As shown in Figure 5A, typical patterns of γ-H2AX in H1299 cells following transduction with C6B were observed by confocal microscopy. In some cells, significant background staining was observed, but that may correspond to the detection of endogenous stress, which is relatively high in H1299 cells. In contrast, no staining could be evidenced with C6BM (Figure 5B and Appendix A), indicating that the monitored signal with C6B is specific. Importantly, γ-H2AX could also be specifically detected in H-treated U2OS cells under these conditions (Appendix A).

To further analyze whether we could use the C6B chromobody in transduction experiments, we produced C6B-mCherry and C6B-dTomato fusion proteins in *E. coli*. dTomato protein was tested because its intrinsic fluorescence brightness is approximately three times higher than that of mCherry [33]. The expected structures of these molecules are schematically depicted in Figure 5C. The analysis on gel of these fusion proteins showed that C6B-mCherry was systematically cleaved during the protein preparation, whereas C6B-dTomato protein (referred hereafter to as C6B-dTo) migrated as a single band (Figure 5D). The delivery in H1299 cells of C6B-dTo led to the specific visualization of foci upon RS induction (Figure 5E, enlarged micrographs). No nuclear signal was observed with C6BM-dTo and the delivered polypeptides remained in this case in the cell cytoplasm and accumulated next to the nuclear membrane (Figure 5E). In addition, we observed a more intense fluorescence signal in the nuclei of transduced cells treated with gemcitabine + AZD-7762 instead of hydroxyurea (Figure 5F). We also confirmed the importance of bivalency of C6B in this context. Transduction of purified C6-dTo proteins under similar conditions did not allow foci detection and the fusion proteins preferentially accumulated in the nucleoli (Appendix A), demonstrating that the specific binding of the bivalent C6 nanobody is maintained within the crowded intracellular context. As a control, we stained the C6B-dTo-transduced cells with mAb 3F4 before microscopic analysis. Notably, the foci detected with C6B-dTo strictly co-localized with those visualized with the antibody and secondary Alexa fluor 488-labelled globulins (Appendix A). Since a similar staining pattern was observed in hydroxyurea-treated U2OS (Appendix A) and in H1299 cells treated with clofarabine or triapine—two other drugs that target the ribonucleotide reductase enzyme as does hydroxyurea (Appendix A)—we concluded that the possibility of specific binding through enforced avidity due to the dimerization of the dTomato protein could represent an added value for the true detection of γ-H2AX in the crowded intranuclear space of mammalian cells.

### 2.9. Real-Time Analysis of γ-H2AX in Drug-Treated H1299 Cells

To follow the fate of γ-H2AX foci in live cells, we took advantage of the strong fluorescence signal emitted by the dTomato protein and the fact that low amounts of C6B-dTo molecules can be delivered in cells via our electroporation method. Preliminary experiments showed that almost all of the internalized molecules accumulated in the nucleus when 0.5 to 2 µg of purified fusion protein were used. Figure 6A shows typical nuclei of C6B-dTo-transduced H1299 cells monitored by wide-field microscopy following treatment of the cells with hydroxyurea or gemcitabine + AZD-7762 for 24 h. Whereas no foci could be observed in the untreated cells, tiny amounts of them were clearly observed after transduction of 0.5 µg of protein. The signal was higher when 2 µg were used and expectedly even more intense when the cells were treated with gemcitabine + AZD-7762 instead of hydroxyurea (Figure 6A). Images were acquired every minute over a period of 10 min (Appendix A). Image processing to subtract the background signal (Materials and Methods) allowed us to distinctly record γ-H2AX foci and their movement over time (Figure 6B, Appendix A). Interestingly, by calculating the trajectories of the foci, we found that those showing bright signal are less mobile that those with low signal. In addition, the foci do not move into the nucleoli and, in some cells, we found that their speed was not homogenous over the whole nucleus (see Figure 6B, lower panel). We have also checked whether γ-H2AX foci can be observed when lowering the time of incubation of the cells following treatment with gemcitabine + AZD-7762 (Appendix A). Whereas most γ-H2AX-positive nuclei displayed individual foci as observed with hydroxyurea, some of them showed the typical pattern of mid-S phase nuclei (Appendix A, lower panel) that has been observed after transfection of cells with a PCNA-GFP construct [34]. Collectively, the data show that γ-H2AX foci can be imaged without ambiguity in live cells after transduction of C6B-dTo, enabling to study the dynamics of this particular histone modification.

### 2.10. Impact of the Delivered C6B Nanobody on Cell Survival

To assess if the delivered C6B nanobody interferes with the cell response to genotoxic drugs, we performed cell survival assays with transduced H1299 cells and monitored the γ-H2AX levels following pulse-treatment with hydroxyurea for 24 h. Cells transduced with either PBS, C6B, or C6B-dTo grew similarly at day 1, 2, and 3 post-treatment with hydroxyurea (Figure 6C) and there was no alteration of the cycling state of the C6B-transduced cells when compared to PBS-transduced cells after drug withdrawal (Appendix A). This suggests that the delivered C6B molecules are not toxic. Furthermore, quantitative Western blotting showed that phosphorylation of H2AX was maximal 24 h after the pulse-treatment and almost undetectable after three days of drug withdrawal (Figure 6D and Appendix A). This correlates well with the regrowth of the transduced cells and indicates that the binding of the C6B nanobody does not interfere with the cell response to hydroxyurea. The fact that C6B appears to be non-neutralizing in the cells, since it does not interfere with the γ-H2AX turnover, makes C6B an extremely powerful tool for imaging.

## 3. Discussion

The detection of γ-H2AX with antibodies in fixed cells is an established method for assessing RS and/or DNA damage. Due to the lack of methods for following this PTM in living cells, it is not clear whether H2AX phosphorylation corresponds to a transient state of the chromatin following damage which allows the coordination of responses and recovery upon injury or if it is a consequence of the spontaneous strong activation of defined kinases, which is finally reverted by the re-start of DNA replication in cycling cells. Notably, the typical structures observed with antibodies after cell fixation can persist over a long time and it has been shown that this pattern might be of relevance for therapeutic purposes [35]. Since upon insult γ-H2AX levels vary from one cell to another, our goal was to obtain a reagent that would allow monitoring in individual cells both γ-H2AX levels and their fate after treatment with varying doses of genotoxic agents. Because classical antibodies are relatively expensive reagents and cannot diffuse freely into the nucleus upon delivery, we have generated nanobodies that can be advantageously produced in bacteria and have the intrinsic property to fold stably in a reducing environment [36]. Their capacity to diffuse into the nucleus makes them usable as intrabodies in cells following either transduction or transfection.

It is well known that generating nanobodies against small peptides or chemicals is very challenging and might be more difficult than against globular proteins [37,38]. The immunogen used here consisted of a small peptide harboring a phospho-serine linked chemically to a carrier protein. This conjugate was found to be adequate for eliciting a strong anti-γ-H2AX response of B cell in mice as described in a previous work [16]. Despite the fact that the titers of the alpaca’s sera were relatively high upon collecting the PBMCs, we had to perform numerous selection attempts to isolate one clone that could specifically interact with the peptide used for immunization, confirming that single-domain antibodies cannot easily bind to small linear epitopes. This might also be the reason why another group was unsuccessful in selecting an anti-γ-H2AX nanobody following immunization of a lama with the same peptide [26]. These results suggest that the main subclass of IgG produced in these animals after immunization with a small peptide are IgG1 molecules bearing a light chain and not the IgG2 and IgG3 HcAbs that are the source of nanobodies. This might be an important consideration for future nanobody generation.

The amino acid composition of the isolated VHH variants differed minimally. Indeed, only two residues in the CDR3 (positions 98 and 99) were altered, whereas the residues in the two other CDRs were conserved. Interestingly, the 3D structure of the VHH-peptide complex showed that these two altered residues are not essential for the interaction with the phosphorylated peptide. It is possible that these alterations reflect the recombination variability that occur during the rearrangement of the V (variable), D (diversity), J (joining) gene segments, which further suggests that the isolated variants originate from a single B cell. Strikingly, the nature of these two residues in CDR3 was found to be critical for the stable folding of the nanobodies upon expression in the bacterial cytoplasm, since clone G2 was almost insoluble compared to C6, for instance (Appendix A). Whether these small changes within the VHH domain also affect the folding in mammalian cells remains to be tested. Moreover, the 3D structure analysis revealed that both the phosphate group carried by S139 and the C-terminal Y play a preponderant role in recognition, which could explain the observed specific binding of the variants to the phosphopeptide. It has been reported that the C-terminal Y of γ-H2AX can also be phosphorylated and this may trigger cell death signaling [11]. Our nanobodies do not bind to a peptide that contains both modifications (i.e., phospho-S139 and phospho-Y142), indicating that γ-H2AX detected in our experiments following drug treatment is not phosphorylated at the C-terminus. Moreover, it is noteworthy to mention that the two contiguous R in CDR3 at positions 100C and 100D that participate actively in the recognition of the γ-H2AX C-terminal end and whose mutation abrogates binding, are also present in the CDR3 of the established anti-γ-H2AX mAb 3F4. This suggests that this conserved arginine motif is of outmost importance for affording specificity.

It is not surprising that it was necessary to render the nanobody bivalent to detect γ-H2AX without ambiguity. Indeed, several groups have reported that having multiple copies of the VHH coding region enhances binding through avidity [36,39]. This strategy enabled the identification of the discrete γ-H2AX foci that are formed early after treatment of the cells. It has been reported that these γ-H2AX foci are arranged into nano-domains, called nano-foci, only visible by super-resolution microscopy [40]. Thus, if the γ-H2AX molecules are in close proximity in the foci, bivalent binding is possible, thereby precluding the fast dissociation, which is instead likely for the monovalent molecules. Previously reported alpaca-derived nanobodies recognizing γ-H2AX functioned well in vitro [26], but none of them was able to specifically interact with γ-H2AX in cells. We speculate that this is due to the fact that these nanobodies were expressed as monovalent molecules, thus, lacking the bivalency important for stable target binding in the cellular context [26]. We proved the binding capacity of the bivalent nanobody to γ-H2AX with different methods, among which treatment of the cells with CSK buffer prior to fixation. This experiment, which was particularly instrumental for the rapid validation in cells of the binding-defective mutant nanobody that served as reliable negative control, demonstrates that our bivalent nanobody is stably bound to γ-H2AX in cells. Nevertheless, we were somewhat surprised to observe that specific binding to γ-H2AX after pulse-treatment with H has almost no impact on the ability of the cells to recover from drug-induced RS. In particular, it seems that the nanobody does not interfere with the cellular response to RS or, more precisely, that the masking of phosphorylated S139 with the nanobody does not hinder the action of phosphatases that allow the injured cells to return to their initial state after drug withdrawal. Whether our nanobody is not of sufficient affinity to counteract these activities remains to be clarified.

To our knowledge, this study shows for the first time that γ-H2AX foci can be specifically detected and tracked in living cells. Our experiments indicate that γ-H2AX is readily accessible in the nucleus, is not buried into the chromatin and is rather free from interactions with MDC-1, contrary to what was proposed by Rajan and colleagues [26]. The images taken after transduction of the purified C6B nanobody into the cells were very similar to those obtained after cell fixation and subsequent antibody incubation, indicating that the localization of γ-H2AX can be scrutinized in the crowded context of live cells in an unbiased way. Interestingly, γ-H2AX was also easily detectable with the chemically-labelled fluorescent C6B nanobody, making it a potential reagent for the quantitative determination in one step of γ-H2AX levels in high-throughput screening assays [10,41]. Although well applicable in vitro, the Alexa Fluor-nanobody conjugates formed aggregates in the transduced cells upon prolonged incubation at 37 °C, which prompted us to test the performance of chromobody proteins delivered into the cells. Even though chromobodies directly expressed in cells after transfection have been successfully applied to monitor several intracellular antigens in live cells [42], their amount in the cells can hardly be controlled and their overexpression generates in general high background staining, especially when the amount of intracellular target is low. Lower expression levels of the chromobodies, achieved either by reducing their half-lives or by adopting weak promoters to drive their expression, can help overcome this issue allowing for the visualization of endogenous antigens with more confidence [43,44]. We were surprised to observe that our nanobody fused to either eGFP or mCherry and expressed in the cells by transfection re-localizes mostly to the nucleus, even in the absence of any drug treatment. Its preferred localization in this compartment upon transfection is likely due to overexpression and unspecific binding.

To overcome this problem, we have implemented a transduction strategy based on the concept of low loading and potential cross-reactivity in the cell. It is widely admitted that antibody-based molecules can bind to different epitopes, albeit with varying affinity, a property termed polyspecific [45]. Our bivalent nanobodies, when fused to dTomato protein, cannot diffuse into the nucleus due to their size (molecular weight of approximately 120 kDa). This is what we observed with the C6BM-dTo molecules, which remain spatially sequestered in “aggresome-like” juxta-nuclear regions [46]. However, when their binding capacity is not abolished, as it is for C6B-dTo, they can recognize sites with somewhat low affinity and when one of these sites is present on a newly synthesized nuclear protein in the cytoplasm, they become piggybacked in the nucleus. In this compartment, due to the on-off rate of the recognition and weak binding at equilibrium, the C6B-dTo molecules are stabilized by the strongest binding reaction [43], even more when the antigen recognized with a higher fit is present in excess. The low nuclear fluorescence signal observed in the untreated cells can be partly explained by the fact that these cells are not under RS and continue to divide, which dilutes the low signal present in the nucleus. By contrast, the stressed cells stop dividing and γ-H2AX levels raise considerably with time. In addition, delivery of sub-saturating amounts of bivalent nanobodies labelled with a fluorescent protein with a high quantum yield was paramount for observing distinctly the foci. Moreover, as schematically depicted in Appendix A, the imaging method described calls on alternated binding events within the cell. It can probably be applied to other nuclear antigens displaying epitopes where the antibody can bind with various degrees of fit. In our case, we suspect that binding to the non-phosphorylated C-terminus of H2AX (that was undetectable in vitro after washing) is involved in the transport of the nanobody into the nucleus. Whether it will be possible to monitor with this system the appearance of the foci and their progressive formation over longer periods of incubation after addition of drug(s) to the cancer cells remains to be investigated. It is likely that such analyses in single living cancer cells will contribute to the knowledge of the chromatin landscape modifications in response to drug treatment [47].

## 4. Materials and Methods

### 4.1. VHH Libraries and Phage Selection

Alpacas (*Llama pacos*) were immunized at days 0, 21, and 35 with the synthetic phosphorylated peptide CKATQA(p)SQEY corresponding to the C-terminus of H2AX after covalent cross-linking to ovalbumin (150 µg). The immunogen was mixed with Freund complete adjuvant for the first immunization and with Freund incomplete adjuvant for the following immunizations. The immune response was monitored by titration of serum samples by ELISA with immunizing peptide on plate. Bound antibodies were detected with anti-alpaca rabbit IgG [48]. For the construction of the libraries, blood samples (200 mL) of the immunized animals were collected under strict veterinary control and the PBMCs were isolated by Ficoll gradient centrifugation (GE Healthcare, Vélizy-Villacoublay, France). For the preparation of total RNA, approximately 10^7^ cells were lysed with the TRIzol reagent (ThermoFisher Scientific, Grand Island, NY, USA). Complementary DNA (cDNA) was amplified using either SuperScript IV reverse transcriptase (ThermoFischer Scientific) or the BD Smart RACE kit (BD Biosciences, San José, CA, USA). The VHH repertoires were amplified from the cDNA by two successive PCR reactions using three different primer pairs (PCR1, PCR2; Appendix A) and the VHH fragments were cloned into the SfiI/NotI restriction sites of the pHEN1 phagemid vector. After transformation into either *E. coli* TG1 or XL1-blue cells by electroporation, the bacterial colonies (approximately 4 × 10^7^ independent transformants per library) were infected with M13KO7 helper phage to produce the phage libraries. The recombinant phages of each library were purified by PEG 8000/NaCl precipitation and aliquots were stored at −80 °C after addition of 15% glycerol. Biopanning was performed with the phosphopeptide (0.5–5 µg/mL) coated on microtiter wells (ThermoFisher Scientific). Briefly, approximately 10^11^ phages in PBS containing 5% nonfat-dried milk were added to uncoated wells for 1 h and subsequently transferred to the peptide-coated wells. After incubation at 20 °C for 1 h, the wells were extensively washed with PBS containing 0.05% Tween 20. Bound phages were eluted with trypsin and amplified in growing TG1 cells for the next round of selection. The amount of phosphopeptide coated on plate was lowered to 0.5 µg/mL in the second round of selection. Phage titers and enrichment after each panning round were determined by infecting TG1 cells with 10-fold serial dilutions of the collected phages and plating on LB agar plates containing 100 µg/mL ampicillin and 1% glucose. Where indicated, binding of the phages to antigen on plate was revealed with an anti-M13 monoclonal antibody conjugated to horse radish peroxidase (HRP; Abcam, Cambridge, UK). The VHH nucleotide sequences were determined using the M13-RP primer (GATC-Eurofins, Ebersberg, Germany).

### 4.2. VHH Engineering and Bacterial Production

The coding region of the selected VHHs in the pHEN1 vector were amplified by PCR with primers VHH-BspHI-Deg and VHH-NotI-short and subcloned into the pET-E6T-6H expression plasmid, a derivative of pETOM [49], which contains in frame at the NotI site the E6 epitope tag recognized by the 4C6 mAb and a His6 tag. To generate the bivalent VHH constructs, the coding region of the VHH was amplified by SOE-PCR with the primer pairs pETOM-For/G4S-Rev and G4S-For/E6T-Rev. The G4S-Rev and G4S-For are the annealing primers to add the (G4S)_3_ linker region. The recombinant fragment was cloned into the NcoI-digested pET-C6-E6T-6H plasmid after digestion with NcoI restriction enzyme, thus, generating pET-C6B-E6T-6H. To generate the C6 mutant construct, which harbors an Ala residue and an Ile residue instead of the two Arg residues at position 100C and 100D in the CDR3 region, we amplified by SOE-PCR the coding region of the C6 with primers pETOM-For and pETOM-Rev, in combination with C6-Mut-Rev and C6-Mut-For as annealing primers. The resulting PCR fragment was sub-cloned into the NcoI/NotI-digested pET-C6-E6T-6H to obtain pET-C6M-E6T-6H. The plasmid pET-C6BM-E6T-6H, which encodes the bivalent form of the mutated C6 coding region, was constructed as described above. The additional Cys residue in the coding region of the bivalent C6 was obtained by amplification of the C6 coding region with primers VHH-BspHI-For and C6-Cys-Rev and sub-cloned into the pET-C6B-E6T-6H.

The pET-C6B-mCherry and pET-C6B-dTomato plasmids were constructed by inserting in frame the coding regions of mCherry protein or dTomato protein in the unique BamHI located in the E6 tag region. The dTomato coding region was subcloned from the ptdTomato-N1 vector (Clontech, Mountain View, CA, USA). All primers used to generate the above-described plasmids are listed in Appendix A.

The VHH variants were expressed in *E. coli* BL21(DE3) plysS cells by addition of IPTG (1 mM) and incubation overnight at 20 °C. The expressed polypeptides were purified as previously described [49], except that IMAC chromatography was performed on a HITRAP^TM^ IMAC HP 1 mL column (GE Healthcare) loaded with nickel ions. The C6 variants with a cysteine residue at the C-terminal end of the second VHH coding region were purified on HIS TRAP^TM^ Excel columns (GE Healthcare) in the presence of 2 mM TCEP. All buffers used in the process were supplemented with 1 mM EDTA and 0.2 mM PMSF. Where indicated, the eluted samples were further purified by size exclusion chromatography on a Superdex 75 10/300 GL column equilibrated in 20 mM HEPES buffer pH 7.2 containing 50 mM NaCl, 1 mM EDTA, 0.1 mM PMSF and 2 mM TCEP (optional). The C6B-mCherry and C6B-dTomato fusion proteins were purified by IMAC chromatography on HITRAP^TM^ columns as above and subsequently polished by size exclusion chromatography on a HILOAD 16/600 Superdex 200 PG column (GE Healthcare) equilibrated in PBS. All purified proteins were stored at −80 °C after addition of 10% glycerol.

### 4.3. ELISA

For the ELISA assays, microtiter wells (ThermoFisher Scientific) were coated with 1 µg/mL of phosphorylated or non-phosphorylated peptide CKATQASQEY in PBS overnight at 4 °C. The purified VHH preparations were diluted in PBS containing 0.2% non-fat died milk and following incubation at RT for 1 h they were revealed with mAb 4C6 and subsequent addition of HRP-conjugated rabbit anti-mouse IgG (GE Healthcare). After several washes with PBS containing 0.1% NP40 and addition of 3′,3′,5′,5′-tetramethylbenzidine (Sigma-Aldrich, St. Louis, MO, USA), the optical density was measured at 450 nm in an ELISA reader. The data were processed with R software using the *drc* package [50].

### 4.4. SPR Analysis

All experiments were performed on a Biacore T200 instrument at 25 °C in HBS-P buffer containing 10 mM HEPES (pH 7.4), 150 mM NaCl, 0.05% P20 surfactant. The phosphopeptide CKATQA(p)SQEY was immobilized on the biosensor surface (BR-1005-30; GE healthcare) through the SH group of the N-terminal cysteine using thiol coupling chemistry. The reference surface was treated similarly except that peptide injection was omitted. The purified VHH samples were serially injected in duplicate for 120 s over reference and peptide surfaces. Each sample injection was followed by a wash with HBS-P buffer during 600 s. Sensorgrams were corrected for signals from the reference flow cell as well as after running buffer injections. The Kd was determined by fitting the equilibrium response (Req) versus the concentration curve to a 1:1 interaction model with the Biacore 2.0.2 evaluation software (GE Healthcare). Responses were normalized relative to phosphopeptide density as fractional occupancy (FO) of target sites [32].

### 4.5. 3D Structure Determination

Purified C6 protein was incubated for 1 h with a 1.3-fold excess of phosphopeptide treated with 2 mM N-ethyl maleimide to prevent dimerization. The complexes were subjected to size exclusion chromatography on a Superdex 75 10/300 GL column (GE Healthcare) equilibrated in 20 mM HEPES buffer pH 7.2, 150 mM NaCl. The peak fractions were concentrated to 5.1 mg/mL with a Amicon Ultra 3K filter (Merck-Millipore, Burlington, MA, USA). The crystallization experiments were carried out by the sitting-drop vapor diffusion method at 20 °C using a Mosquito Crystal dispensing robot (TTP Labtech) for mixing equal volumes (200 nL) of the C6-peptide sample and reservoir solutions in 96-well 2-drop MRC crystallization plates (Molecular Dimensions). Crystallization conditions were tested using commercially available screens (Qiagen, Molecular Dimensions, Hilden, Germany). Several wells were found positive after about one week of incubation and crystals obtained with 25% PEG 3350, 0.2 M sodium acetate. The crystals were transferred to 35% PEG 3350, 0.2M sodium acetate before being flash cooled in liquid nitrogen. The data were collected at the Proxima 2A beamline of the synchrotron Soleil at a wavelength 0.98 Å (12.65 keV) on an EIGER X 9M detector (Dectris) with 20% transmission. 360° of data were collected using 0.1° oscillation and 0.025 s exposure per image, with a crystal to detector distance of 134.25 mm. The data were indexed, integrated, and scaled using XDS [51]. The 3D structure of the C6/phosphopeptide complex was solved by molecular replacement using the PHASER module of PHENIX [52] with the structure of VHH PorM_01 (PDB ID: 5LZ0) edited to remove water molecules and the CDR loops, being used as a search model. Refinement was performed using the refine module of PHENIX [52] followed by iterative model building in COOT [53]. The structural figures were prepared with Chimera-X software (http://www.rbvi.ucsf.edu/chimerax, accessed on 29 March 2020).

### 4.6. Cell Culture, Transduction, Histone Preparation, Western Blotting and FACS Analysis

The H1299 and U2OS cells (laboratory stocks) were maintained in Dulbecco’s modified Eagle’s tissue culture medium (DMEM; Life Technologies, Carlsbad, CA, USA) supplemented with L-glutamine (2 mM), gentamicin (50 µg/mL) and 10% heat inactivated fetal calf serum at 37 °C in a humidified 5% CO_2_ atmosphere. Fresh cells were thawed from frozen stocks after 10 passages and mycoplasma contamination was tested by DAPI staining. Counting of the cells was performed with the automated cell counter LUNA-II (Logos Biosystems, Villeneuve d’Ascq, France). Where indicated, the cells were treated with either hydroxyurea (H; 2 mM), gemcitabine (G; 0.1 µM), AZD-7762 (A; 0.1 µM), clofarabine (C; 0.3 µM), triapine (T; 2 µM), camptothecin (CPT, 1 µM), epirubicin (EPI, 0.5 µM), etoposide (ETO, 10 µM), cisplatin (CIS, 10 µM), oxaliplatin (OXA, 10 µM), or combinations of two drugs at the same concentration as indicated. The drug concentrations used are those described in a previous work [16]. All drugs were purchased from Sigma-Aldrich.

Transduction experiments with purified C6, C6B, C6B-dTomato proteins or Fab 3F4 were performed essentially as previously described [30]. Briefly, 10^5^ cells in PBS were mixed with the protein sample (0.5–2 µg) and subjected to electroporation (1550 V, 10 msec, 3 pulses) using the Neon transfection device (Life Technologies). The treated cells were incubated for 1 h at 37 Q°C in medium and, after centrifugation for 5 min at 100× *g*, the pelleted cells were seeded and allowed to recover overnight in complete medium without antibiotics before addition of the drugs.

For the purification of the histone proteins, the harvested cells (approximately 10^7^/mL) were lysed for 10 min at 4 °C in PBS supplemented with 0.5% Triton X100, 2 mM PMSF, 0.02% NaN_3_ and 1 mM Na_3_VO_4_. After centrifugation for 10 min at 6500× *g* at 4 °C, the recovered nuclei were acid extracted overnight at 4 °C in 0.2 M HCl. The histone proteins present in the clarified lysate were stored at −20 °C.

For the analysis of the H1299 proteins by Western blotting, soluble extracts (60 μg/lane) in RIPA buffer were used. γ-H2AX and β-actin were revealed with monoclonal antibody 3F4 (0.1 μg/mL) and rabbit polyclonal serum A2066 (Sigma-Aldrich), respectively. Bound secondary HRP-labeled antibodies were revealed with ECL reagent (GE Healthcare, Chicago, IL, USA). The intensity of the relevant signals was quantified with an Image QuantLAS 4000 imager (GE Healthcare, Chicago, IL, USA) using manufacturer’s software.

Analysis of the transduced cells by FACS was performed essentially as previously described [54]. Briefly, the cells were fixed on ice with cold 70% ethanol for 1 h. After rehydratation, the cells were treated with RNaseA (10 µg/mL) and stained with PI (3 µM) (ThermoFisher Scientific). Analysis of the cycling state was performed with a Accuri C6+^TM^ flow cytometer (BD Life Sciences, San Jose, CA, USA) using manufacturer’s software.

### 4.7. Construction of the pβ-Actin Plasmids and Transient Transfection

The pβA-scFv-eGFP, a derivative of pDRIVE-hβ-actin [55], was modified by PCR to insert in frame to the scFv the E6 tag and the mCherry protein using E6T-HindIII-For/E6T-HindIII-Rev and mCherry-For/mCherry-Rev primer pairs, respectively. This vector, which carries unique NcoI and SpeI restriction sites, was used to sub-clone the VHH variants as described above, thereby generating pβA-C6-E6T-mCherry, pβA-C6M-E6T-mCherry, pβA-C6B-E6T-mCherry, and pβA-C6BM-E6T-mCherry. All oligonucleotides used to construct these expression vectors are listed in Appendix A.

The day before transfection, 8 × 10^4^ cells were plated in 12-well culture plates containing glass coverslips. Transient DNA transfection was performed using jetPRIME (Polyplus Transfection, Illkirch, France) according to manufacturer’s instructions. The culture medium was replaced with fresh medium after 4–24 h of incubation with the polymer/plasmid mixtures. Cells were incubated (37 °C, 5% CO_2_) for 40 h (hydroxyurea-treated cells) or 24 h (gemcitabine + AZD-7762-treated cells), followed by microscopic analysis.

### 4.8. Immunofluorescence Microscopies

For the analysis by classical immunofluorescence microscopy, the transfected or transduced cells were fixed with 4% paraformaldehyde for 20 min and, after permeabilization with 0.2% Triton X100 for 5 min, they were incubated with mAb 3F4 or VHH preparations diluted in PBS containing either 10% fetal calf serum or 2% BSA. Where indicated, the cells were treated with CSK-100 modified buffer (100 mM NaCl, 300 mM sucrose, 3 mM MgCl_2_, 10 mM HEPES pH 6.8, 1 mM EGTA, and 0.2% Triton X-100) for 5 min prior to fixation. The VHH molecules were revealed by addition of mAb 4C6, which binds to the E6 tag and bound antibodies were detected with Alexa Fluor 488 or 568 labelled-anti-mouse immunoglobulins (Life Technologies). Where indicated, Alexa 568 labelled-C6B molecules were used. The labelling of the purified bivalent C6 equipped with a cysteine residue at the C-terminus was performed essentially as previously described [30]. Briefly, purified protein in 0.1 M KH_2_PO_4_ pH 6.5, 150 mM NaCl, 1 mM EDTA, 250 mM sucrose was mixed with a 1.2 molar amount of Alexa Fluor 568 maleimide derivative (ThermoFisher Scientific). After adjustment of the pH at 7.5 and subsequent incubation for 1 h at room temperature, the chemical reaction was blocked with N-ethylmaleimide in excess. The mixture was centrifuged through either a Zeba spin column with a cut-off of 7 kDa (GE Healthcare) or the fluorescent dye removal column provided by Thermofischer Scientific. The amount of fluorophore per bivalent C6 in the flow-through was calculated by spectrophotometry with a Nanodrop 2000 device (ThermoFisher Scientific). After incubation of the cells with the different reagents and several washes with PBS, the coverslips were mounted with 4′,6′-diamino-2phenyl-indole (DAPI) Fluoromount-G (Southern Biotech, Birmingham, AL, USA) and imaged with a Leica DM5500 microscope (Leica, Wetzlar, Germany) equipped with 20X and 63X objectives. The signal was recorded with a Leica DFC350FX camera. Confocal microscopy was performed as previously described [21]. All microscopy images were processed using the Fiji/Image J software. For the measurement of the nuclear fluorescence intensity, the images of microscopy fields were acquired with the 20X objective. The nuclei were set with the DAPI channel acquisition as regions of interest (ROI) and the mean fluorescence intensity in each ROI was measured using the Fiji built-in-tool and data were processed with the R software.

Widefield fluorescence microscopy was performed on a home-built system composed by a Nikon TiE inverted microscope coupled to a high-numerical aperture (NA) TIRF objective (Apo TIRF 100X, oil, NA 1.49, Nikon). Live-sample were illuminated with a laser diode at 561 nm (10 W/cm^2^, Oxius) at 37 °C. Real-time imaging was performed by introducing a single edge dichroic mirror and a bandpass filter in the emission path of the microscope (Semrock, 560 nm edge BrightLine single-edge imaging-flat dichroic beamsplitter, 593/40 nm BrightLine single-band bandpass filter) and by using an EM-CCD camera (ImagEM, Hamamatsu, 0.106 µm pixel size) with a typical integration time of 100 ms. The videos were recorded using the perfect focus system of the microscope to avoid z-drift during the acquisition (1 image recorded every minute during 10 min). Images were processed using Fiji. To visualize the movement of the foci, we used a filtering procedure in which two different Gaussian blurs (A = 1.3 pixel and B = 2 pixels) were applied to each image of the stack. The improved stack was obtained by computing the difference between A and B. The Mosaic plugin was then used on the final stack to reconstruct the single foci trajectories over the whole acquisition.

### 4.9. Statistical Analysis

Statistical analysis was performed using R software version 3.6.1. Averages are represented as means +/−SD and the number of replicates is indicated in the figure legends. In the boxplots (Figure 2, Figure 3, Figure 4 and Figure 5), the bars indicate the median and interquartile range of the recorded fluorescence after processing with R software. The statistical significance of the data obtained after transfection (Figure 3) was determined with the Student’s *t*-test and indicated as *** *p*-value < 0.001. Prior to the Student’s *t*-test, normality and equality of variances were tested using Shapiro–Wilk’s test and Fisher’s *F*-test, respectively. For the correlation analysis (Figure 4), normality of the data was tested using Shapiro–Wilk’s test and correlation was evidenced by calculating the Pearson’s correlation coefficient.

## 5. Conclusions

The novel nanobody presented here is useful for analyzing γ-H2AX in various formats of DNA damage assays. In particular, this tool might be of interest for the establishment of rapid screening assays of chemicals for genotoxicity. Because it can be easily engineered and produced in *E. coli*, it represents a cost-effective reagent when compared to a validated conventional antibody. Moreover, this nanobody and the transduction strategy described will likely pave the way to reliably study the dynamics of γ-H2AX in individual cancer cells under a variety of conditions and in response to genotoxic stimuli in particular. Finally, we believe that this approach can be applied to the tracking of numerous other nuclear proteins and PTMs for the study of RS in real time in cancer cells.

## 6. Patent

Université de Strasbourg. Single domain antibody specific for phosphorylated H2AX and its uses. EP21305627.8; 12 May 2021.

## Figures and Tables

**Figure 1 cancers-13-03317-f001:**
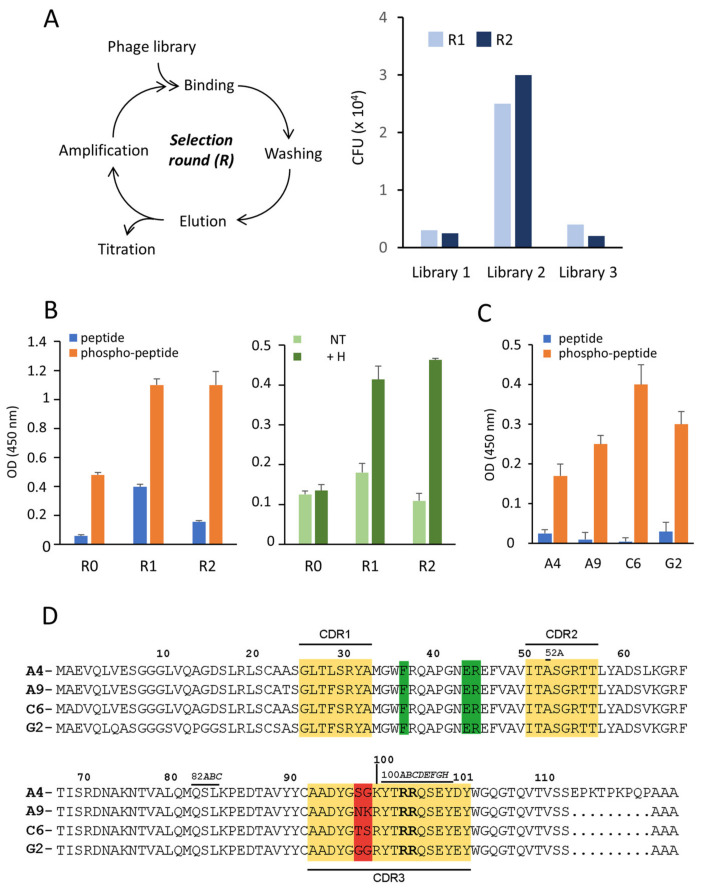
Development and selection of specific anti-γ-H2AX nanobodies. (**A**) Schematic representation of a phage display selection round (left). The histogram on the right shows the number of phages retained on plate after two rounds of selection with three different libraries issued from PBMCs of individual animals. CFU, colony-forming units. (**B**) The specific binding capacity of the phages selected from the library 2 were assayed by phage-ELISA with either peptides as indicated (left) or histones extracted from H-treated or untreated (NT) cells (right), both immobilized on plate. (**C**) Four individual VHH-phages identified by sequencing (A4, A9, C6, and G2) were subjected to phage-ELISA. Their specific binding to either the non-phosphorylated (peptide) or the phosphorylated (phosphopeptide) H2AX C-terminal peptide is shown. Bound phages were revealed with an HRP-labelled anti-M13 conjugate (Materials and Methods). (**D**) The sequences of clones A4, A9, C6, and G2 are aligned based on homology, according to the Kabat numbering system [27]. The residues highlighted in yellow correspond to the CDR residues. Residues 114 to 125 are part of the hinge region. The R residues at positions 100C and 100D are shown in bold and the residues highlighted in green are hallmark residues of the VHH variable domain [28].

**Figure 2 cancers-13-03317-f002:**
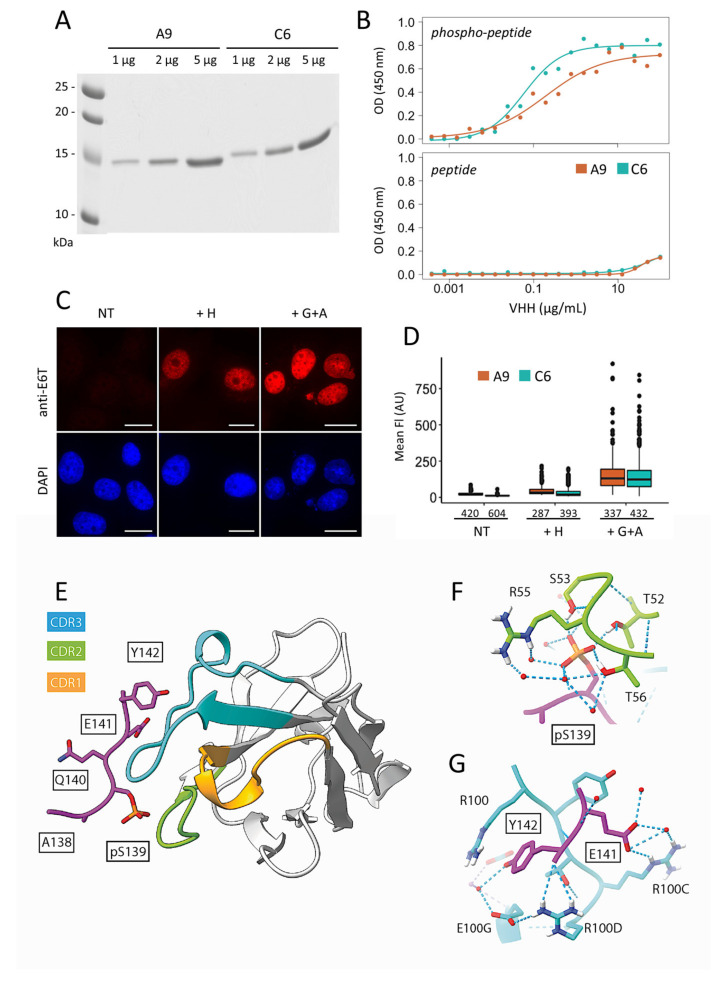
Biochemical and structural analysis of the selected nanobodies. (**A**) SDS-PAGE analysis of the purified nanobodies A9 and C6. (**B**) The binding capacity of the purified samples shown in (**A**) was tested by ELISA with either phosphorylated (phosphopeptide; 1 µg/mL) or non-phosphorylated (peptide; 1 µg/mL) C-terminal H2AX peptide coated on plate. (**C**) Immunofluorescence assay with C6 nanobody. H1299 cells were treated for 24 h with the indicated drugs (H, hydroxyurea; G+A, gemcitabine + AZD-7762) and incubated after fixation with nanobody C6. Bound molecules were revealed with anti-tag E6 antibodies (anti-E6T) and Alexa 568-labelled anti-mouse IgG. The nuclei were counterstained with DAPI (blue). Scale bar: 20 µm. NT, untreated cells. (**D**) Quantification of the γ-H2AX fluorescence signal recorded following incubation of the cells treated as in (**C**) with either A9 or C6 nanobody. (**E**) Crystallographic 3D-structure of the C6 nanobody in complex with the phosphorylated peptide corresponding to γ-H2AX C-terminal tail (ApSQEY). The CDR1, CDR2, and CDR3 loops colored in yellow, green, and blue, respectively. The γ-H2AX tail is shown in magenta and peptide residues are boxed. (**F**,**G**) Close-up view of the γ-H2AX tail peptide in the nanobody binding site. Residues are color coded and labelled as in (**E**). Water molecules in the interface between the γ-H2AX tail and the nanobody are represented as red spheres and hydrogen bonds are represented as blue dotted lines.

**Figure 3 cancers-13-03317-f003:**
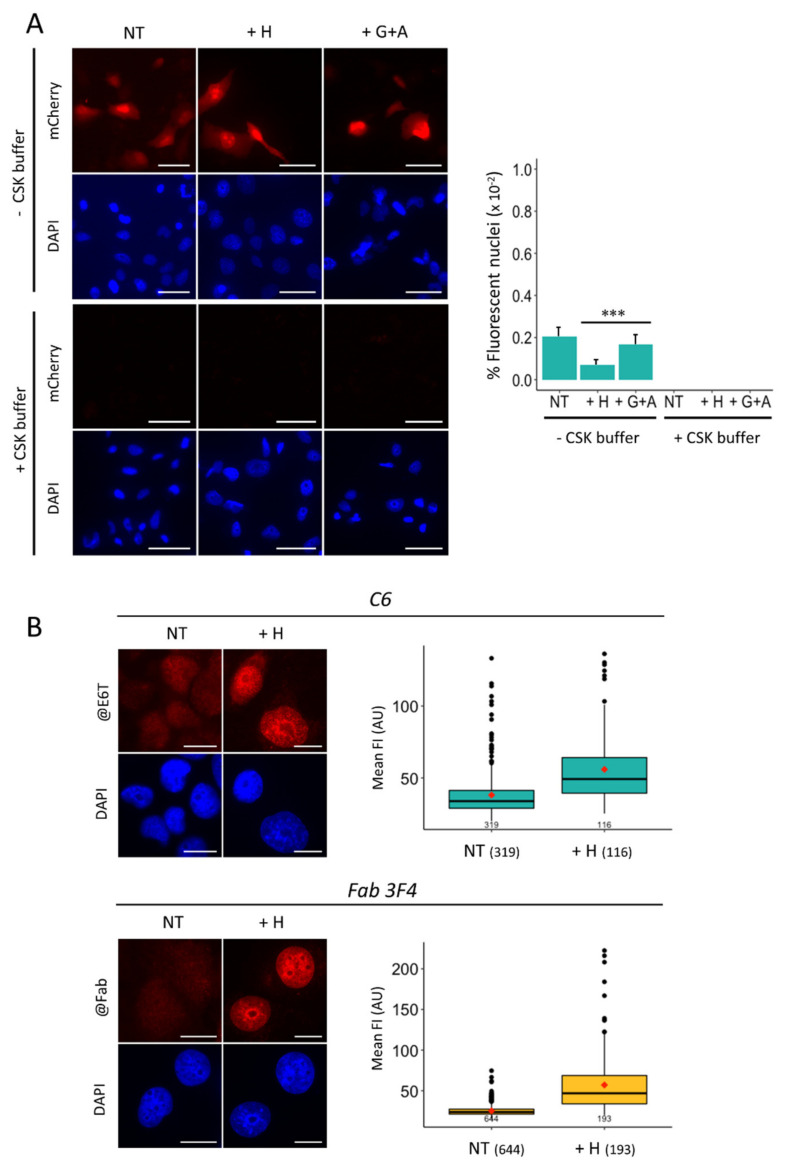
C6 nanobody localization to the nucleus in drug-treated H1299 cells. (**A**) Immunofluorescence analysis of H1299 cells after transfection with the plasmid encoding the C6-mCherry chromobody. 24 h post-transfection, the indicated drugs (H, hydroxyurea; G+A, gemcitabine + AZD-7762) were added. After incubation for 24 h, the cells were either fixed (- CSK buffer) or washed with CSK buffer prior to fixation (+ CSK buffer). The nuclei were counterstained with DAPI (blue). The panel shows representative images recorded under the microscope (left) and the percentage of fluorescent cells observed in each condition is shown (right). Cut-off for negative cells was set on non-transfected cells using the maximal recorded value. Data are represented as mean ± SEM, *** *p* < 0.001. Scale bar: 50 µm. NT, untreated cells. (**B**) Following transduction with either C6 nanobody or Fab 3F4, H1299 cells were treated with H and analyzed by immunofluorescence 48 h post-treatment. Representative images are shown on the left. Bound nanobody or Fab were revealed with anti-E6 tag antibodies (anti-E6T) and secondary Alexa 568-labelled anti-mouse globulins (anti-IgG). Scale bar: 20 µm. The quantification of the γ-H2AX mean fluorescence intensity (FI) of the monitored cells is shown on the right. The numbers indicated in brackets correspond to the number of cells analyzed in each condition.

**Figure 4 cancers-13-03317-f004:**
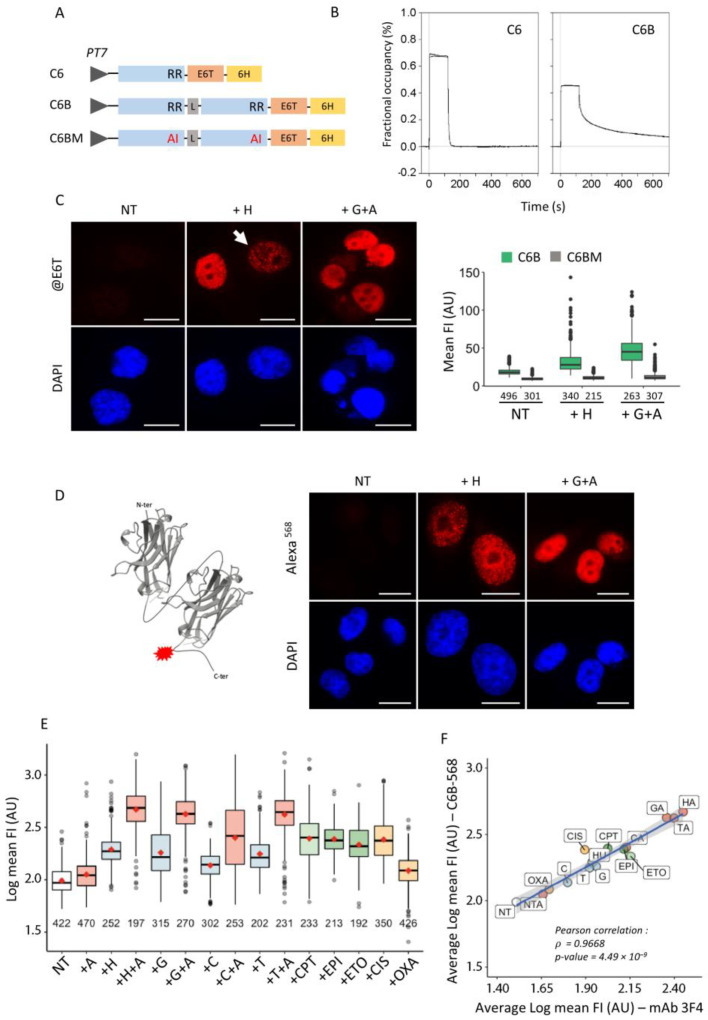
Binding performance of the bivalent C6 nanobody. (**A**) Schematic representation of the constructs used to produce bivalent nanobodies in *E. coli* cells. The four R residues of bivalent C6 nanobody (C6B) that have been altered to generate the mutant bivalent C6 nanobody (C6BM) are indicated. (**B**) Analysis by SPR of the interaction of monovalent (C6; 180 nM) or bivalent (C6B; 80 nM) C6 nanobody with the phosphopeptide immobilized on chip. The curves show typical normalized profiles of the fractional occupancy calculated with the signals recorded for each nanobody (Materials and Methods). Injection of nanobody was stopped at the 120 s time-point and dissociation was analyzed during 700 s. (**C**) Representative immunofluorescence images of hydroxyurea (H)- or gemcitabine + AZD-7762 (G + A)-treated H1299 cells following fixation and incubation with bivalent C6 nanobody (left). Bound material was revealed as described in the legend of Figure 3. The nuclei were counterstained with DAPI (blue). Scale bar: 20 µm. The quantification of the mean γ-H2AX fluorescence intensity (FI) of these analyzed cells and those monitored after incubation with the C6BM molecules is shown on the right. The number of cells analyzed in each condition is indicated. NT, untreated cells. (**D**) Detection of γ-H2AX in drug-treated H1299 cells as in (**C**) with the fluorescently-labelled C6B. A depiction of the bivalent nanobody chemically conjugated to Alexa 568 (red) is shown on the left. An immunofluorescence analysis of drug-treated H1299 cells after incubation with the labelled conjugate is shown on the right. Nuclei were counterstained with DAPI (blue). Scale bar: 20 µm. (**E**) Box plot representation as in (**C**) of the normalized γ-H2AX FI detected with the C6B-Alexa 568 conjugate of H1299 cells after treatment with the indicated drugs or drug combinations. The data shown correspond to those recorded after log transformation. The full name and the concentration of the drugs used is indicated in the Materials and Methods section. The numbers indicated in the x axis correspond to the number of cells analyzed in each condition. NT, non-treated. (**F**) Comparison of the C6B-Alexa 568 conjugate with the mAb 3F4 for detecting γ-H2AX in drug-treated H1299 cells. The FI data obtained with cells treated as in (**E**) and incubated with mAb 3F4 and Alexa 568-labelled secondary antibodies were plotted against the data shown in (**E**). The means in each case were taken to generate the curve. The calculated Pearson correlation coefficient is indicated.

**Figure 5 cancers-13-03317-f005:**
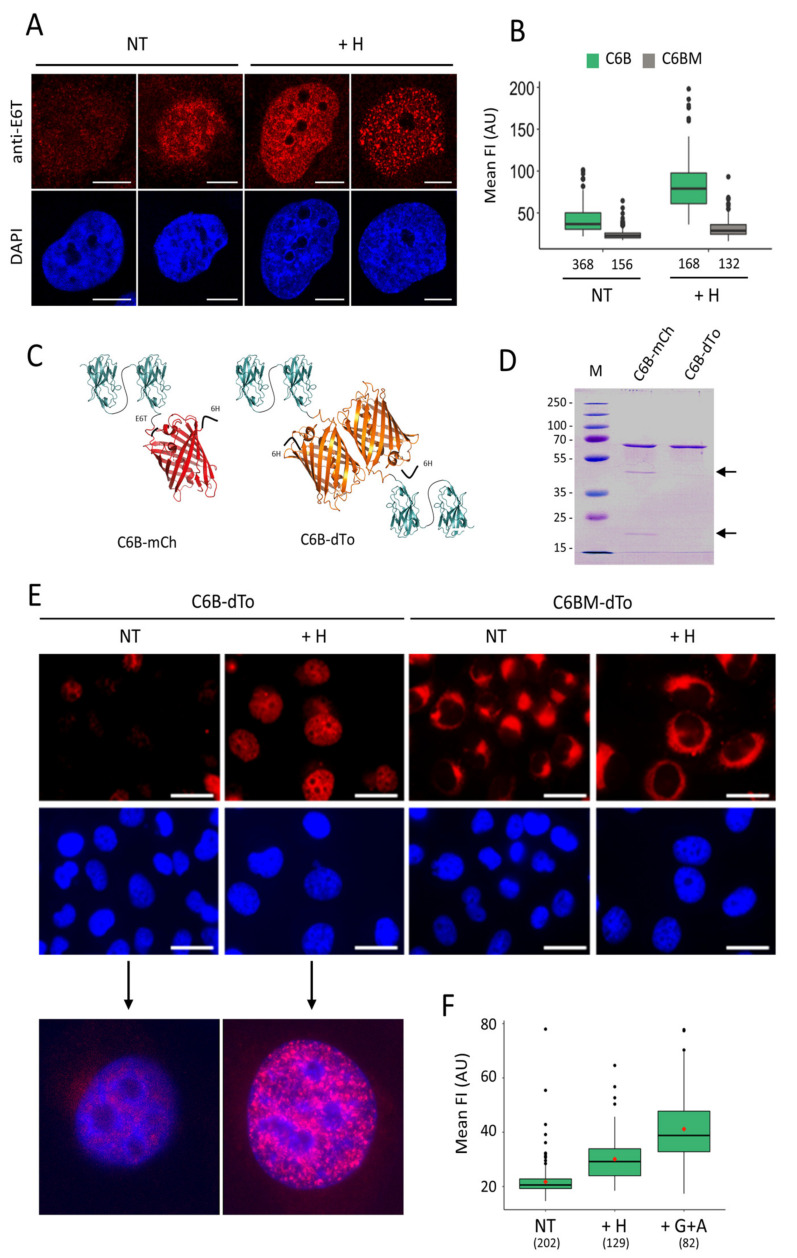
Detection of γ-H2AX with the bivalent nanobody upon delivery by electroporation. (**A**) H1299 cells transduced with either C6B or C6BM nanobodies were treated with hydroxyurea (H) and revealed with anti-tag E6 antibody (anti-E6T) and Alexa 568-labelled anti-mouse globulins 40 h post-treatment. Typical immunofluorescence images of C6B-transduced cells taken with a confocal microscope after DAPI counterstaining (blue) are shown. Scale bar: 10 µm. (**B**) The quantification of the mean FI of cells transduced as in (**A**) with either C6B (green) or C6BM (grey) are represented. The number of analyzed cells in each condition is indicated (bottom). (**C**) Schematic representation of the C6B-mCherry (C6B-mCh) and the C6B-dTomato (C6B-dTo) fusion proteins used in the study. (**D**) Analysis by SDS-PAGE of the purified C6B-mCh and the C6B-dTo fusion proteins. The proteolytic products observed in the C6B-mCh samples are indicated with arrows. M, molecular weight markers (kDa). (**E**) Analysis by immunofluorescence microscopy of the C6B-dTomato (C6B-dTo) and C6BM-dTomato (C6BM-dTo) fusion proteins (red) following transduction in H1299 cells. 24 h post-transduction, the cells were treated with hydroxyurea (H) or left untreated (NT). The images show typical fields observed in each case under the microscope after fixation and DAPI counterstaining (blue). Scale bar: 20 µm. An enlargement of one cell present in the field of the C6B-dTo samples following overlay of the red and blue channels with Fiji is shown below the original images. (**F**) The quantification of the nuclear mean FI of C6B-dTo-transduced H1299 cells after treatment with the drugs is shown. The numbers at the bottom correspond to the number of cells analyzed in each case.

**Figure 6 cancers-13-03317-f006:**
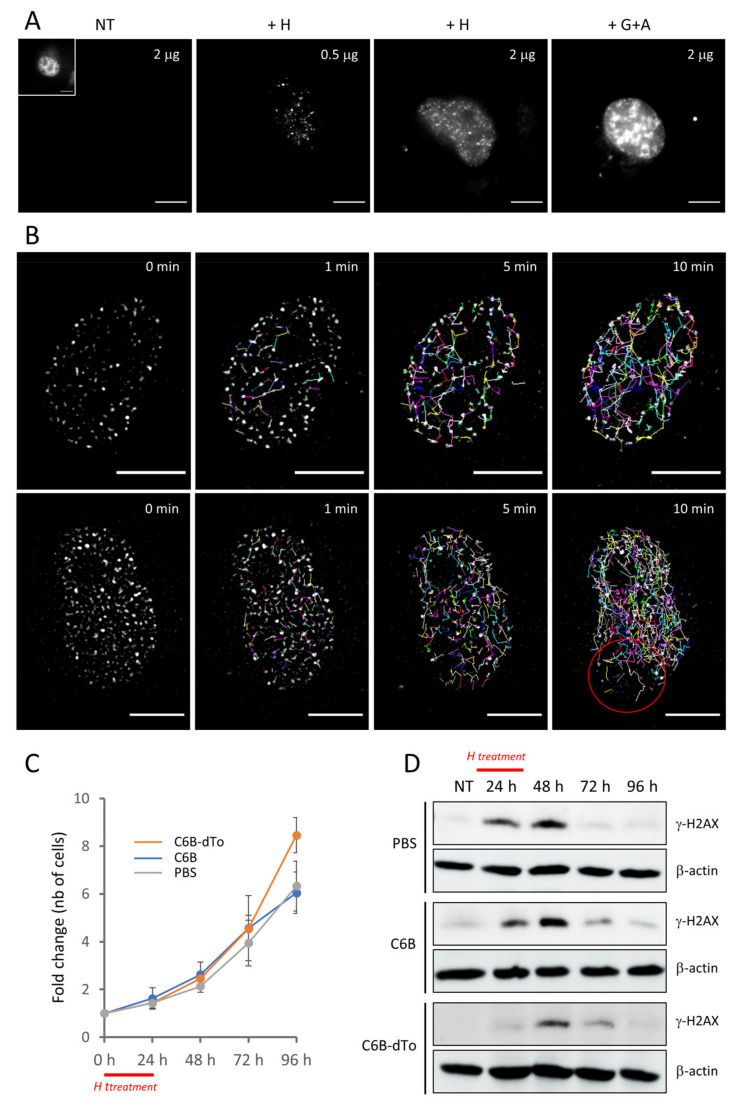
Visualization of the binding of the bivalent nanobody in live H1299 cells and analysis of its effect after pulse treatment with hydroxyurea. (**A**) Representative wide-field fluorescence microscopy images of H1299 cells transduced with the C6B-dTomato fusion protein and subsequently treated with the indicated drugs (H, hydroxyurea; G+A, gemcitabine + AZD-7762) or left untreated (NT). Images with an identical exposure time were taken 24 h after treatment. The amount of protein used for the transduction in each case is also indicated. Scale bar: 10 µm. The nucleus shown in the inset correspond to the nucleus of the NT panel after 8-fold enhancement of the exposure time. (**B**) Analysis of the movement of the foci formed in C6B-dTo-transduced H1299 cells treated with hydroxyurea (H). 24 h post-treatment, the cells were analyzed as in (**A**) and pictures were taken every minute (total time: 10 min). The recorded images were processed as indicated in the Materials and Methods section and show the trajectories of the foci present in two typical cells after 0, 1, 5, and 10 min of incubation. Scale bar: 10 µm. (**C**) Growth rate of the transduced H1299 following pulse-treatment with H. After transduction with the indicated proteins, the cells were seeded on plate and pulse-treated during 24 h with H. The curves correspond the number of cells after counting at 24, 48, 72, and 96 h after seeding. The data correspond to the calculated ratios (number of cells in each case/number of cells at seeding time (0 h)). (**D**) Variation of γ-H2AX levels in H1299 cells transduced with either PBS or the C6B or C6B-dTo proteins and treated for 24 h (pulse treatment) with hydroxyurea as probed by Western blotting with mAb 3F4. Following treatment, the transduced cells (5 × 10^5^) were incubated in fresh medium and extracts (50 μg) were prepared at the indicated time points. β-actin was used as a loading control. NT, extract of untreated cells (before hydroxyurea treatment). The original blots are shown in Appendix A.

## Data Availability

The data of the 3D structure of the C6 nanobody have been deposited to the PDB (PDB ID 6ZWK). Additional data related to this paper may be requested from E.W.

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
