# Peer review of "A Novel Nanobody Precisely Visualizes Phosphorylated Histone H2AX in Living Cancer Cells under Drug-Induced Replication Stress"

_cancers, 2021, doi:10.3390/cancers13133317_

Round 1
Reviewer 1 Report
Overall, this article contributes to the advancement of the nanobodies and chromobodies technology that detect γH2AX in living cells.
I would like to ask the following questions and make a few suggestions that would improve this review article:
- Have you developed or attempted to develop a not phosphorylated peptide? That would also be a valuable control. Please, speculate on this.
- Why you have not used irradiated cells? Irradiation with X-rays generates more defined foci. Please, speculate on this.
- It should be indicated which inhibitor of the ck1 is used in the experiments.
- All the acronyms should be explained in the figures. For example, in Fig.2, the NT, H, +C+A, @E6T are not explained in the legend. Some labelling is missing in Fig. 5
- In Fig.5 (E), it seems there is a difference in staining pattern between the NT and +H in C6BM-dTo samples. Do you have an explanation for this?
- The following references should be added to the bibliography section of the paper: DNA double-stranded breaks induce histone H2AX phosphorylation on serine 139
E P Rogakou, D R Pilch, A H Orr, V S Ivanova, W M Bonner doi: 10.1074/jbc.273.10.5858
Megabase Chromatin Domains Involved in DNA Double-Strand Breaks in Vivo
Emmy P. Rogakou, Chye Boon, Christophe Redon, and William M. Bonnera doi: 10.1083/jcb.146.5.905
Author Response
Response to Reviewer 1:
We thank the Reviewer for their suggestions.
- Our aim was to develop a nanobody that specifically binds to the phosphorylated C-terminal end of H2AX. During the phage selection, we have used the non-phosphorylated peptide linked to beads to perform a counter-selection. Basically, we eliminated phages binding to the non-phosphorylated peptide, which allowed the selection of nanobodies in which several CDR residues interact directly with the phosphate group of the phosphorylated S139, as shown in Figure 2E. In addition, we have confirmed by ELISA that the two well-expressed nanobodies do not bind to the non-phosphorylated peptide (Figure 2B). Throughout the manuscript, we also used untreated cells as control to show that our nanobodies do not recognize the non-phosphorylated form of H2AX present in the untreated cell nuclei. Together, we have used the non-phosphorylated C-terminus of H2AX as a control in almost all presented in vitro and in cellulo Finally, we would like to mention that we have also performed two phage-display selection experiments with the non-phosphorylated peptide as bait on beads, and did not observe any significant increase in the number of phages retained on beads after the second and third round of panning.
- We agree that X-rays would generate more focused foci, which would be interesting to analyse with our nanobody. However, at the moment, we do not have access to X-ray sources and would depend on external collaborators to perform such experiments. However, establishing new collaborations requires time, and this would delay the availability of the tool to the scientific community.
- The name of the Chk-1 inhibitor used in the study is AZD-7762. This is indicated in the Materials and Methods section. We have also “AZD-7762” in the text (line 199).
- We thank the Reviewer for this suggestion. Indeed, this would certainly be helpful for the readers We have added the full name of all drugs used for the described experiments in the main text and in the relevant figure legends. We have also replaced “@E6T” by “anti-E6T”. E6T is the abbreviation for E6 tag.
- The Reviewer is right: the staining pattern of the C6BM-dTo molecules in the non-treated cells is not identical to that observed in hydroxyurea(H)-treated cells. We speculate that the difference is due to the swelling of the nuclei upon H-treatment, which can be clearly observed by DAPI staining. Nonetheless, importantly, the signal is cytoplasmic in both cases, indicating that the mutant nanobody does not recognize the antigen. We generally observe swelling of the nuclei following treatment of cancer cells with replication inhibitors for 24 hours (less so with topoisomerase inhibitors). It has also been observed by others and it could be related to RS-related chromatin de-condensation and/or induction of senescence, but there’s no clear explanation until now for this drug-induced phenotypic change.
- We have added the two suggested references describing these important findings about the formation of g-H2AX upon DNA double-strand breakage.
Reviewer 2 Report
Moeglin et al successfully developed an nanobody to detect and visualize gamma-H2AX by using phage display, and characterized the nanobodies in vitro and in cells. This study provides a useful tool to the field, which is important. The study is well designed and paper is well written.
Minor:
I would suggest not to use abbreviations in main text, such as H, A+G.
Font size of line 503 is not correct.
Author Response
Response to Reviewer 2:
We were happy to read that the Reviewer found our paper well written, the study well designed and the tool useful to the community. We thank the reviewer for the comments and suggestions.
- In the main text, we have replaced all abbreviations with the full words. In particular, we have exchanged H with hydroxyurea and G+A with gemcitabine + AZD-7762.
- We changed the font size of line 503.
Reviewer 3 Report
Development of a nanobody to specifically visualize histone H2AX phosphorylation in fixed and living cancer cells under drug-induced replication stress
Overall comments
Moeglin et al. demonstrated a couple of potentially exciting data detecting gamma-H2AX foci, one of the most reliable biomarkers of double-stranded DNA breaks, in living cells. This reviewer approves the potential significance and novelty of this study. The major concerns are listed below, which this reviewer would urge the authors to consider for revision, if appropriate, to help convey the paper’s central theme. Also, several minor concerns, which would help the readers’ understanding, are listed as well.
Major concerns
In this paper, several chemicals are frequently used to provoke replicative stress (RS), such as H, G, and A, by which fluorescence images and/or mean values of FI are demonstrated in all figures. However, surprisingly, no experiment is designed and presented in this paper demonstrating a dose-dependent linearity effect, by which the concentration of each chemical used here would be judged something appropriate. If the primary purpose of this paper is to develop a new reagent for detecting something important (such as kinetics of gamma-H2AX in living cells without fixation), which nobody could detect so far, scientifically reproducible, quantitative, and rigorous results must be demonstrated. To this end, this reviewer firmly believes that quantification of each result would be a key parameter to judge the significance and novelty of the discovery. This reviewer agrees that replicative stress under optimal culture conditions is adequate to generate gamma-H2AX, which can be routinely detectable by a standard anti-gamma-H2AX antibody, although the cells have to be fixed. If the primary goal of this paper is to develop an alternative reagent to detect gamma-H2AX but in living cells, why should the authors use RS-inducible chemicals? The authors should show us side-by-side the results of immunofluorescence microscopic analysis of gamma-H2AX foci in serum-stimulated vs. serum-starved cells by both the standard gamma-H2AX antibody and the nanobody. After serum starvation for 24-48 h, gamma-H2AX by RS is going to be almost zero. Then, add 10% serum (such as FBS) and harvest them for detecting gamma-H2AX foci at 0, 2, 4, 8, and 12 h.
This time-course experiment will give us a chance to judge how sensitive the abovementioned reagent (nanobody) is for detecting “natural” RS-dependent gamma-H2AX foci compared with the standard anti-gamma-H2AX. Even if not as sensitive as the standard gamma-H2AX immunofluorescence analysis, this reviewer believes that the nanobody methods would still be worth pursuing because of the advantage of detecting gamma-H2AX in living cells. Lastly, if there is a chance for revision, please send the supplementary materials in PDF. This reviewer was not able to access the materials.
Minor concerns
Line 2-3: Title: The main title is the shortest abstract. This reviewer would suggest modifying the original title slightly. For example, “A novel nanobody precisely visualizes phosphorylated histone H2AX in living cancer cells under replication stress” or something like this. A shorter title is likely to be preferable as far as it is catchy and striking.
Line 100: “γ-H2AX phosphorylation” should be changed to “γ-H2AX generation”. γ-H2AX is already phosphorylated at Ser139.
Line 102: “the levels of H2AX” would be more appropriate to be changed to “the rate (or efficiency) of gamma-H2AX formation”. The word “level(s)” usually indicates the “amount(s)” of (in many cases) a protein.
Line 134: What species? Human H2AX? Or, did the authors choose the aa peptide sequence that shows high homology among several species, including mouse, rat, monkey, and zebrafish?
Line 145: “…only a low fraction of the cloned …” What does ‘low’ means? Low molecular weight? Low concentration? Please clarify it.
Line 160: What H2AX phosphorylation are the authors talking about here? S139?, Y142? Or T136? Please clarify it.
Line 172: What does “anti-@-H2AX mean”? Correct it to anti-γ-H2AX. Line 172-173: Fig. 1A: What does CFU stand for? Explain it in the legend.
Line 175: “H-treated….” Indeed, in line 159, the authors gave us the complete spelling of H for hydroxyurea, which is better than nothing, but this brief explanation only in the main text may force the readers to check back the main text again particularly when s/he tried to interpret the Figures. “H” itself does not generally represent “hydroxyurea.” The authors do not have to
change the Figure itself but add this key in the Figure legend. H: hydroxyurea. This suggestion is also applicable for Fig. 2 and line 199. Please add the keys in the Fig. 2 legend: G: gemcitabine, A: chk-1 inhibitor. Please remember that the purpose of having a Figure legend is to help the readers to understand the Figure(s) smoothly.
Line 613: What is the size (or length) of a VHH fragment? A phage display library usually limits the size of each peptide to a certain length.
Line 180: Any reference for the “Kabat numbering system”?
Line 181: Why not residue 102 and residue 103, instead of 100C and 100D, respectively?
Line 181: The Fig. 1 legend says that “residues 114 to 125 are part of the hinge region.” However, there is no residue 125 in the Figure. 1D. Within the same Fig. 1D, the distances between residue 80-90, 90-100, and 100-110 do not seem to be consistent.
Line 182: Only one amino acid is a hallmark residue of the VHH variable “domain”? This explanation is hard to follow. It would help if the authors showed us the 2nd and 3rd conserved residues and structurally conserved regions. Please also cite a relevant reference here.
Line 201: always lowest? There are only two players; one is A9, and the other is C6. So, this sentence should be changed to “C6’s fluorescence image (FI) value was always lower than that of A9” or something like that (see next).
Line 201: The abovementioned correction should be considered only if the difference of FI between A9 and C6 is statistically meaningful or significant. However, there is no P-value, so this reviewer would judge that the difference is not statistically significant (N.S.)
Line 208: In Fig. 2C, there is no key to explain what @E6T stands for? Does this mean “anti- E6T antibody”?
Line 278: Fig. 3A (right panel). This Figure seems a little strange. Does it mean that only 0.2% of cell nuclei show fluorescence images? In Fig. 3B, the authors should not use laboratory
jargon, such as @. Instead of @, they should use “anti-.” Also, the authors should add a P-value in there.
Line 321: Fig. 4E: Once again, please give us the P-value.
Author Response
Response to Reviewer 3:
We thank the Reviewer for the thorough analysis of our paper and for the constructive criticism. We have performed new experiments to address all concerns.
Major concerns:
- dose-dependent linearity effect of the drugs used in the study (first half of the first paragraph)
We have extensively characterized the dose-dependent genotoxicity of hydroxyurea or gemcitabine + chk-1 inhibitor or other combinations with the mAb 3F4 in a previous work (Moeglin et al., Cancers, 2019). Here, we have used the optimized concentrations that allow observing either moderate or intense replication stress (RS) after 24 hours of incubation for testing the performance of the selected bivalent nanobody. The drug doses and the duration of the treatment used in this study are roughly the same as those used in numerous other publications to show the specific formation of g-H2AX under these RS conditions and the linearity effect of the drugs as well. The goal of this study was not to test the efficacy of the drugs used, but to show that our nanobody is useful for detecting drug-induced g-H2AX in living cells, something that cannot be done with an anti-g-H2AX antibody. We have added the sentence “The drug concentrations used are those described in a previous work (Moeglin et al., 2019)” in the Materials and Methods section.
Nonetheless, to confirm that our tool is adequate for detecting the dose-dependent effect of hydroxyurea, we have performed an additional immunofluorescence assay after treatment of H1299 cells with varying concentrations of hydroxyurea (below). The results show that our nanobody can be used to monitor the gradual increase of g-H2AX levels upon treatment of the cells with increasing concentrations of hydroxyurea (Figure 1).
Figure 1: Detection of g-H2AX in H1299 cells treated with increasing amounts of hydroxyurea. The amount of added drug is indicated on top. g-H2AX was revealed with either C6B nanobody and anti-E6 tag IgG globulins (C6B; upper panels) or with mAb 3F4 and anti-IgG globulins (3F4; lower panels). The analysed cells were counterstained with DAPI (blue). Scale bar: 20 mm. NT, untreated cells.
- detectability of endogenous replication stress (called “natural” replication stress in the comments)
Cancer cells are naturally stressed due to their fast division rate and it has been shown the DDR counteracts the toxic effects of endogenous RS. We are interested to monitor drug-induced RS in living cells and have thus focused our work on the detection of g-H2AX upon drug treatment. The detection of endogenous stress (which cannot be clearly observed in our NT samples by conventional microscopy) would require a very sensitive camera. Moreover, it still would be difficult to ascertain if the signal detected in the absence of drug treatment truly corresponds to endogenous g-H2AX instead of being unspecific binding. Even commercially-available anti-g-H2AX antibodies do also react with H2AX, albeit at a lower rate than to g-H2AX, thus the signal could be due to cross-reaction with H2AX or another protein.
This is probably why the Reviewer proposed to analyse the g-H2AX levels in cells after serum starvation. As asked by the Reviewer, we have tried to synchronize H1299 cells by the removal of serum in the culture medium for 2 days and tested whether g-H2AX protein can be specifically evidenced upon addition of serum to the cells. We did not observe any significant difference between the seemingly arrested cells and those fed with complete medium for 2, 4, 6 or 24 hours after serum starvation (Figure 2, below). Probably H1299 cells are not adequate for performing this experiment since they looked rather sick under the microscope after 2-3 days of serum starvation.
Figure 2: Quantification of the mean FI of H1299 cells after 2 days of serum starvation (0 h) and subsequent addition of complete medium for 2, 4, 6 or 24 hours. The number of analyzed cells in each condition is indicated (bottom of the boxes).
To try to answer the Reviewer’s question, we have analysed non-treated cells by immunofluorescence to monitor endogenous g-H2AX, using both the validated mAb 3F4 and the C6B nanobody (Figure 3, below). In both cases, we found only few cells showing a signal and, in these cells, this was very faint. As written above, we cannot claim that the detected foci correspond to the specific staining of phosphorylated H2AX foci under these conditions.
Figure 3: Detection of endogenous g-H2AX in non-treated H1299 cells. After fixation, the cells were incubated with either nanobody C6B or mAb 3F4. Bound material was revealed as described in the legend of Figure 1. The exposure time of these pictures is 1.2 sec (two times more than for those shown in Figure 1). Scale bar: 20 mm.
Finally, we think that the detection of endogenous g-H2AX formation is beyond the scope of this work and this is why we have added the words “under drug-induced replication stress” to the title. We have developed this nanobody tool to track drug-induced RS in living cancer cells, which, to our knowledge, is not achievable with conventional anti-g-H2AX antibodies.
Minor concerns:
Line 2-3: we have followed the suggestion and have simplified the title. Nevertheless, we kept “under drug-induced replication stress” since, as discussed above, the experiment with serum starvation did not work in our hands.
Line 100: we thank the Reviewer for spotting this. We have made the suggested change.
Line 102: we have replaced “the dynamic changes in the levels of g-H2AX during the treatment” with “the dynamic changes of g-H2AX formation during the treatment”.
Line 134: we have added the word “human”.
Line 145: we have replaced the word “fraction” with the word “number”.
Line 160: we have added “at Ser139” to clarify.
Line 172 and line 172-173: the typo has been corrected and we have added in the legend: “CFU, colony-forming units”. The M13-based phages can easily be converted into colonies on plate. This is the universal method for titrating the phage preparations. Here, we have used this method to count the recovered peptide-binding phages upon elution.
Line 175: we agree with the Reviewer and are grateful for the suggestion. We have replaced all H and G+A abbreviations with “hydroxyurea and gemcitabine + AZD-7762”, respectively, in the main text. Also, we now explain in all relevant legends what is meant by either “H” of G+A” in the Figure.
Line 613: as shown in Figure 1D, the average length of a VHH is 113 residues. Indeed, phage display has been initially developed for selecting short peptides of 10-20 residues, but a number of authors have now shown that Fab molecules (MW: 50 kDa) can be displayed on the tip of the M13 phage. In previous studies, we have extensively experienced the selection of single-chain Fv fragments by phage display and these molecules are approximately 2 times bigger than VHHs (average number of residues: 220).
Line 180 and Line 181: we have added in the legend of Figure 1 the reference for the Kabat numbering. Since the number of residues of the CDR3 is varying, Kabat et al. have chosen to add a letter to all residues that are not conserved within a sub-family of variable chains. We have modified Figure 1D to clarify the numbering of the residues of our VHHs and the illustration of the sequences and their numbering according to Kabat et al., 1991.
Line 182: the 3 residues that are highly conserved in the VHH FR2 regions and considered as hallmark residues are highlighted in green in the new Figure 1D. A relevant reference (Muyldermans, 2013) has been added in the legend.
Line 201: we agree with the Reviewer and have changed the phrasing of the sentence according to Reviewer’s suggestion. We did not add P values because this was a qualitative observation that does not impact the significance of the results presented in Figure 2D.
Line 208: We used “@” for “anti-“. We have replaced in all figures “@-E6T” with “anti-E6T” and explained in the corresponding legends that it refers to the analysis with the anti-E6 tag antibody.
Line 278: we have quantified the results shown in Figure 3A (left) by measuring the percentage of fluorescent nuclei in each case (Figure 3A, right). In this figure, the percentages are expressed as fractions x 10-2 (indicated in the ordinate). The number 0.2 for instance means 20 and so on. Concerning Figure 3B, we have removed all symbols, as written above (line 208).
Line 321: A large number of our results are based on microscopic observations and it is difficult in this case to show images with more that 2-3 nuclei per field to see the g-H2AX staining distinctly. To prove that the cells shown in each case are not “unique” and correspond to those of several fields, we systematically performed the analysis of the nuclei fluorescence intensity of several hundreds cells in each case with the Fiji software. These analyses are represented as box plots representing all analysed cells. These plots display the variation of the samples of a statistical population, the sample median and the first and third quartiles. We think that these representations are of sufficient robustness to convince the readers that we have not chosen the cells presented in the pictures of the immunofluorescence microscopy experiments. The same is true for Figure 2D, Figure 3B, Figure 4 E, Figure 5B, Figure 5F and also for several Figures in the Supplementary Materials.

Reviewer 4 Report
This manuscript presents the use of a novel nanobody for analyzing gamma-H2AX accumulation in cancer cells. The work is extremely solid and well described. The bilavent version of this nanobody can detect gamma-H2AX in a single-step assay with the same sensitivity as with validated antibodies. More importantly, fluorescent nanobody-dTomato fusion proteins can be used in a transduction strategy to visualize with precision gamma-H2AX foci present in intact living cells following drug treatments. This strategy could be used to perform fast screenings of genotoxic drugs and to study the dynamics of this chromatin modification in individual cancer cells under a variety of conditions. Is this potential application (extensible to other modifications) which makes this study of interest for cancer research. Indeed, pioneering studies (recently confirmed by evidence) showed that epigenetic heterogeneity of cancer cells contributes to therapeutic resistance by multiple mechanisms and subpopulations of phenotypically different cells (persisters) provide pools for selecting drug-resistant mutants. It is therefore crucial to develop strategies to analyze chromatin landscape modifications in response to drug treatments in single living cells. On the other hand, it is crucial to determine if the system of analysis can interfere with the cell response to genotoxic drugs. Here I found the experimental analysis not completely satisfactory (see below).
Major points:
-To assess if the delivered C6B nanobody interferes with the cell response to genotoxic drugs, the Authors performed cell survival assays with transduced H1299 cells and monitored the gamma-H2AX levels following pulse-treatment with H for 24 hours. I think that a cytofluorimetric analysis is also needed to exclude eventual significant changes in the cell-cycle dynamics in the cells transduced with the C6B-dTomato fusion protein and subsequently treated with the indicated drugs.
- I have also some concerns about the analysis of variation of γ-H2AX levels in H1299 cells transduced with either PBS or with C6B or C6B-dTo proteins and treated for 24 hours (pulse treatment) with H as probed by Western blotting with mAb 491 3F4. The Authors claim that the results indicate that the binding of the C6B nanobody does not interfere with the cell response to H but the intensity of the signal and the kinetics of γ-H2AX accumulation in C6B-dTo treated cells appear slightly different from what is observed in PBS-treated cells. Since this is a crucial point, the experiment should be repeated and quantitative data should be shown for at least three independent experiments.
- Previous studies (i.e. Brock et al., 2009; Hinohara et al., 2018; many others) showed that epigenetic heterogeneity of cancer cells contributes to therapeutic resistance by multiple mechanisms and subpopulations of phenotypically different cells (persisters) provide pools for selecting drug-resistant mutants. This view suggests that it is crucial to develop strategies to analyze chromatin landscape modifications in response to drug treatment in single living cells giving strength to the Authors’ approach and should be discussed.
Minor points:
- In fig.1 the legend should mention what Alpaca1, Alpaca2 and Alpaca3 represent
- The Authors propose that C6B-dTo molecules after delivery in the cytoplasm by electroporation bind to newly-synthesized nuclear proteins and are subsequently piggybacked in the nucleus. What is the reason for this indirect mechanism? Why not to add a Nuclear Localization Signal to the construct?
Author Response
Response to Reviewer 4:
We were happy to read that the Reviewer found our work “extremely solid and well described”. We thank the Reviewer for the interesting comments and suggestions for experiments, which we have performed to address the concerns.
Major points:
- Cytofluorimetric analysis of the transduced cells following pulse-treatment
As asked, we have repeated the pulse-treatment experiment after transduction of H1299 cells with either PBS or C6B and instead of counting the cells after drug withdrawal we have subjected them to FACS analysis after fixation and staining with propidium iodide (PI). As shown in the new Supplementary Figure 8E, no significant variation of the cell counts with regard to the DNA content (cycling state) was observed 24 hours post-treatment, confirming that the nanobody has no effect on cell recovery after drug-induced RS. We could not use the C6B-dTo-transduced cells for this assay because the emission wavelengths of PI and dTomato are in the same range. We have added the following sentence “and there was no alteration of the cycling state of the C6B-transduced cells when compared to PBS-transduced cells after drug withdrawal (Figure S8E)” in the main text. We have also added a paragraph in the Materials and Methods section 4.6 describing the protocol used.
- Variation of the g-H2AX levels in C6B-dTo-treated cells in comparison to PBS-treated cells
We do not think that the accumulation of g-H2AX in the extracts of C6B-dTo-treated cells is significantly different to that obtained with PBS-treated cells (2 experiments were done already). The slight variation mentioned by the Reviewer is not reproducible. As asked, we have performed an additional experiment and have calculated the pixel numbers of the area of all relevant bands from the 3 independent experiments done. The g-H2AX values were divided by those of b-actin in each lane and normalized to be able to compare the 3 experiments. The histogram, shown in new Supplementary Figure 8E, indicates that there is no significant difference in the variation of the g-H2AX levels in cells transduced with either PBS, C6B or C6B-dTo after pulse-treatment with hydroxyurea. We have replaced the sentence “Furthermore, Western blotting showed that phosphorylation of H2AX was maximal 24 hours after the pulse-treatment and almost undetectable after 2 days of drug withdrawal (Figure 6D)” with “Furthermore, quantitative Western blotting showed that phosphorylation of H2AX was maximal 24 hours after the pulse-treatment and almost undetectable after 3 days of drug withdrawal (Figure 6D and Figure S8D)” in the main text. The software used for the quantification of the intensity of the bands is now indicated in the Materials and Methods section.
- Application to the study of chromatin landscape modifications in response to drug treatment in single living cells
This is a very interesting comment. Indeed, with our tool, it will be possible to analyse the g-H2AX content and turnover in single living cells in response to drug treatment and this approach will certainly contribute to the knowledge of the chromatin landscape modifications under these conditions. In particular, we are interested to investigate if, within a population of cultured cancer cells, some have a higher ability to resist or are less sensitive to drug-induced stress. Whether this property is related, at least in part, to the rate of g-H2AX formation with regard to the global epigenetic heterogeneity of cancer cells can now be studied. We have added a sentence at the end of the Discussion section to speculate on this. The suggested reference (Hinohana et al., 2018) has also been added.
Minor points:
- Meaning of Alpaca1, Alpaca2 and Alpaca 3 in Figure 1.
We have replaced the words “Alpaca 1, alpaca 2, alpaca 3” by “Library 1, library 2, library 3”, respectively, to clarify (1 library per animal).
- transport of the C6B-dTo molecules in the nucleus
We have shown in previous publications that antibodies can be delivered to the nucleus upon transfer in the cytoplasm by electroporation and binding to newly-synthesized nuclear antigen in the same compartment (Freund et al., mAbs, 2013; Conic et al., JCB, 2018). The same strategy has been used here since the nanobody-dTomato fusions do not passively diffuse into the nucleus. The main advantage of this strategy, when compared to NLS-equipped binders, is that only antigen-bound molecules reach the nucleus, allowing the visualization of the antigen in this compartment. On the other hand, the NLS-mediated transport is extremely efficient. However, in this case, all molecules present in the cytoplasm will be transported to the nucleus and it is difficult to see then under these conditions those specifically bound to nuclear antigen. In other words, the strategy of piggybacking to the nucleus allows filtering out the fraction of the non-binders, which generates background signal since no washing step is possible.
Round 2
Reviewer 4 Report
I am satisfied with the Authors' answers and the changes introduced in the manuscript